# Satellite Data Reveals a Common Combustion Emission Pathway for Major Cities in China

Wenfu Tang[1,*], Avelino F. Arellano[1,*], Benjamin Gaubert[2], Kazuyuki Miyazaki[3], and Helen M. Worden[2]

[1] Department of Hydrology and Atmospheric Sciences, University of Arizona, Tucson, AZ, 85721
[2] National Center for Atmospheric Research, Atmospheric Chemistry Observations and Modeling Laboratory, Boulder, CO 80301
[3] Japan Agency for Marine-Earth Science and Technology, Yokohama, Japan
[*]These authors contributed equally to this work.

*Correspondence to*: Wenfu Tang (wenfutang@email.arizona.edu)

**Abstract.** Extensive fossil fuel combustion in rapidly-developing cities severely affects air quality and public health. We report observational evidence of decadal changes in the efficiency, and cleanness of bulk combustion over large cities in mainland China. In order to estimate the trends in enhancement ratios of CO and $SO_2$ to $NO_2$ ($\Delta CO/\Delta NO_2$ and $\Delta SO_2/\Delta NO_2$) and infer emergent bulk combustion properties over these cities, we combine air quality retrievals from widely used satellite instruments across 2005-2014. We present results for four Chinese cities (Shenyang, Beijing, Shanghai, and Shenzhen) representing four levels of urban development. Our results show a robust coherent progression of declining-to-growing $\Delta CO/\Delta NO_2$ relative to 2005 (-5.4±0.7%/year to +8.3±3.1%/year), and slowly-declining $\Delta SO_2/\Delta NO_2$ (-6.0±1.0%/year to -3.4±1.0%/year) across the four cities. The coherent progression we found is not evident in the trends of emission ratios reported in Representative Concentration Pathway (RCP8.5) inventory. This progression is likely due to a shift towards cleaner combustion from industrial and residential sectors in Shanghai and Shenzhen that is not yet seen in Shenyang and Beijing. This overall trend is presently obfuscated by China's still relatively higher dependence on coal. Such progression is well-correlated with economic development, and traces a common emission pathway that resembles evolution of air pollution in more developed cities. Our results highlight the utility of augmenting observing and modeling capabilities by exploiting enhancement ratios in constraining the time variation of emission ratios in current inventories. As cities and/or countries continue to socioeconomically develop, the ability to monitor combustion efficiency and effectiveness of pollution control becomes increasingly important in assessing sustainable control strategies.

## 1 Introduction

Urban agglomeration, particularly megacities (i.e., cities with >10 million inhabitants), are expected to continue growing (in size and number) over the coming decades (Jalkanen, 2012; World Bank, 2015). Anthropogenic activities are most intense in megacities, accompanied by immense energy consumption mainly in the form of fossil fuel combustion (Mage et al., 1996; Kennedy et al., 2015). These lead to enhanced emissions of air pollutants, greenhouse gases, and waste energy, largely impacting air

quality (AQ), climate, and ecosystems (Baklanov et al., 2016, Lelieveld et al., 2015). At present, estimates of city-to-national-scale emissions from fossil fuel combustion remain uncertain, especially in rapidly-developing regions where combustion is still poorly characterized due to the lack of detailed information on energy use, combustion practices, and pollution control strategies (Streets et al., 2013; Creutzig et al.,
2015). This is also confounded by larger uncertainties on other sources of pollution that may be associated with urbanization (e.g., deforestation, agriculture, and fires). These alone preclude us to accurately assess the changes in atmospheric composition due to anthropogenic activities at scales that are relevant to AQ, energy, and environmental policy (National Academies of Sciences, Engineering, and Medicine, 2016).

      Such is the case for cities in China even with the scientific attention the country has received in
the past decades. As China grew into the world's second largest economy, its rapid development resulted to substantial emissions (Richter et al., 2005), and more frequent occurrences of most severe pollution events in many of its megacities, most notably Beijing (Guo et al., 2014). These affect not only local AQ and public health but are reported to impact hemispheric-to-global atmospheric environment (Lin et al., 2014; Verstraeten et al., 2015). Along with the growth of these cities is a growing body of evidence of
decreasing emissions and associated pollution levels in some cities in China. This points to important changes in AQ as a result of development, AQ management, and regional-to-national socioeconomic initiatives embodied within its Five-Year Plans (FYP) (Reuter et al., 2014; Krotkov et al., 2016; van der A et al., 2017; Sun et al., 2018; Koukouli et al., 2018). However, these changes in AQ as a result of efforts to control air pollution are still obfuscated at present by the increase in combustion activities, along with
uncertainties in bottom-up emission inventories, and diversity in economic structure and growth across cities (Wang and Hao, 2012; Mi et al., 2017). Monitoring these reductions at city scale remains to be a challenge especially when narrowly viewed within the context of a single pollutant, and more so when attributing them to a particular emission sector.

      Fossil fuel emissions from an evolving megacity follow a pattern that can be potentially monitored
and refined, by combining observational constraints on combustion activity (abundance of combustion products) with efficiency and effectiveness of pollution control strategies or 'cleanness' (enhancement ratios of these products) (Silva et al., 2013; Hassler et al., 2016; Silva and Arellano, 2017; Tang et al., 2018, 2019), alongside information on the state of socio-economic development (e.g., gross domestic product (GDP) or income) and *a priori* estimates from bottom-up emission inventories. In particular, the
'cleanness' of combustion of a known fossil fuel type can be determined stoichiometrically by measuring the relative abundance of intermediate products such as carbon monoxide (CO), nitrogen oxides ($NO_X$), sulfur dioxides ($SO_2$), and soot particles with final products like carbon dioxide ($CO_2$). Please see Methods section for more details. Most of these products are currently monitored as criteria pollutants by surface measurement networks and as tracers of pollution by satellite remote sensing (Streets et al., 2013; Duncan
et al., 2014). In fact, these combustion products are revealed in space as very distinct bulk enhancements over a megacity metropolitan location in marked spatial contrast with the city's surroundings (Bechle et al., 2011; Lamsal et al., 2013). At a scale of a megacity being monitored from space, these enhancements are analogous to smoke plumes coming from a stationary smokestack. And so, observations of these megacity plumes enable us to monitor bulk anthropogenic activity and transboundary pollution. They
have also been used in recent years to refine the spatiotemporal distribution of emissions (Lamsal et al., 2013; Hakkarainen et al., 2016; Ding et al., 2017), to indicate bulk combustion efficiency, inter-megacity differences and fire phase (Silva et al., 2013; Silva and Arellano, 2017; Tang and Arellano, 2017), and to

infer fossil fuel $CO_2$ emissions (Konovalov et al., 2016) among others. From an annual to decadal standpoint, it is reasonable to interpret the long-term changes in spatial covariations between these observed pollutant enhancements within the megacity to reflect dominant shifts in bulk combustion characteristics (e.g., changes in fuel mixture and technology practice), which can then be indicative of an emission pathway for a given megacity (e.g., Parrish et al., 2002; Parrish, 2006; Russell et al., 2012; Silva et al., 2013; Hassler et al., 2016; Silva and Arellano, 2017). Data sampling and collocation issues, as well as retrieval information content and chemical nonlinearities between these pollutants, do not quite manifest at decadal scales more than emission changes, especially when treated as a smokestack in the analysis.

In this study, our goal is to uncover space-based evidence of dominant shifts in the cleanness of bulk combustion of large cities across the recent decade (through these ratios), associate these shifts to particular sectors, and identify a common emission pathway across these cities. Along the same line to studies on environmental Kuznets curves (EKC, Stern, 2004) and human development (Lamb et al., 2014), we attempt to connect this pathway to economic growth by finding a power law relationship between the ratios observed for each major city in China and the city's GDP per capita. As cities in China grow, emissions from fossil fuel combustion evolve accordingly depending on the rate and type of socioeconomic development, technological innovation, and environmental policies (Chan and Yao, 2008; Bechle et al., 2011; Zhang et al., 2012; Wang et al., 2012; He and Wang, 2012; Luo et al., 2014; Koukouli et al., 2018; Sun et al., 2018). This evolution however cannot be reflected at shorter time scales. As a basis for comparison, pollution controls adopted in developed countries like United States and Europe, which followed a progression from first controlling $SO_2$, CO, and then $NO_X$ (Crippa et al., 2016), reflect some aspects of decadal-scale sustainable development that can be brought to light in the case of China.

We analyze the emergent patterns of the 'cleanness' of bulk combustion in the past decade (2005-2014), based on enhancement ratios between intermediate products of combustion ($\Delta CO/\Delta NO_2$ and $\Delta SO_2/\Delta NO_2$) observed within each megacity and urban agglomeration in China. We use gridded monthly-averaged satellite retrievals of total columns of CO from Measurement of Pollution In The Troposphere (MOPITT), tropospheric columns of $NO_2$ from Ozone Monitoring Instrument (OMI), and planetary boundary layer (PBL) columns of $SO_2$ from OMI to derive monthly estimates of these ratios. We conduct spatial regression analysis and subsequently derive estimates of the decadal trends of these ratios using time series analysis. We then compare these trend estimates to inferred trends from a couple of model-derived abundance ratios and several emission ratios from current bottom-up emission inventories, including estimates based on the Representative Concentration Pathways scenario (RCP8.5) (Riahi et al., 2011). We also conducted a simple inverse analysis to update the contribution of major emission sectors in RCP8.5 to fit our estimates of decadal changes in enhancement ratios. Section 2 describes data and methods used in this study. Results and discussions are presented in Section 3. Section 4 is summary and implication of this study.

## 2       Data and Methods

### 2.1       Study Region

We considered all 31 provincial capitals and five special cities (Beijing, Shanghai, Shenzhen, Tianjin, and Chongqing) in mainland China for our analysis. These cities comprise the main urban agglomerations in the country (see Figure 1 for coverage). For purposes of finding long-term emergent patterns on its emission characteristics, we focused our analysis to 12 representative urban agglomerations. These 12 cities cover the four economic regions of China (i.e., East Coast: Beijing, Tianjin, Shanghai, Guangzhou, Shenzhen; Central China: Wuhan, Northeast China: Harbin, Shenyang; and Western China: Chengdu, Chongqing, Xian, Hohhot). Based on prior information from RCP8.5 and National Bureau of Statistics of China (http://data.stats.gov.cn), these cities already exhibit largely diverse pollution and economic development attributes illustrated in Figure 1 as differences in magnitude, sectoral, and temporal distribution of emissions and GDP per capita for 2005 to 2014 between these cities. Our goal is to assess whether the long-term patterns that are seen in these *a priori* emission estimates are consistent with observations. We also considered Los Angeles and other large cities in the United States (New York City, Chicago, Houston, Phoenix, Boston, Seattle, and Miami) for comparison.

### 2.2       Data

The main datasets used in this study are summarized in Table 1. This includes multiple satellite retrievals, representative emission inventories, and a couple of model simulations and chemical reanalysis.

### 2.2.1 Satellite Retrievals and Data Processing

We use the NASA Terra Measurement of Pollution In The Tropophere (MOPITT) version 6, Level 2, multispectral (Thermal Infrared/Near Infrared) retrievals of carbon monoxide (CO) total columns for CO (Deeter et al., 2014), tropospheric column retrievals from NASA Aura/ Dutch Ozone Monitoring Instrument $NO_2$ (DOMINO) v2.0 for $NO_2$ (Boersma et al., 2011), and Ozone Monitoring Instrument (OMI) Planetary Boundary Layer (PBL) $SO_2$, version 3, Level 2 (Krotkov et al., 2006). We collected daily MOPITT CO, OMI $NO_2$, and OMI $SO_2$ retrievals that are available within a $2° \times 2°$ area around each city center. This radius was selected to cover the extent of each city based on $NO_2$ footprints (Bechle et al., 2011; Lamsal et al., 2013) and geopolitical maps of city boundaries. We grid each set of retrievals into $0.1° \times 0.1°$ grids that commensurate to the finest retrieval resolution among MOPITT and OMI. We then average them across each month to minimize spatiotemporal collocation issues (see Table 1 for differences in sampling of MOPITT and OMI). As a result, there are 400 points for each species (CO, $SO_2$, $NO_2$) per city and month. We note that these retrievals have been used in the past to study decadal changes for individual (or a pair of) pollutants but not to derive enhancement ratios (e.g., Krotkov et al., 2016). While CO retrieved from thermal infrared (TIR) radiances are mostly sensitive to free tropospheric CO, it has also been reported to be capable of observing lower tropospheric CO, especially when retrieved jointly from TIR and near infrared (NIR) radiances (Worden et al., 2010; Deeter et al., 2014). We recognize however that retrievals of $SO_2$ from OMI have been reported to exhibit low sensitivity to weak

SO$_2$ signals, in particular to less than 30 to 70 kTon per year of point source emissions (Krotkov et al., 2016). While our spatial and temporal smoothing, along with anchoring our SO$_2$ analysis with NO$_2$ data (please see later description of our regression analysis), should help in enhancing the SO$_2$ signal from cities with low SO$_2$ emissions, these SO$_2$ retrievals are useful as large SO$_2$ abundances are still observed across the majority of cities in China (Krotkov et al., 2016). We also used CO retrievals from the Infrared Atmospheric Sounding Interferometer (IASI), Level 2 (De Wachter et al., 2012), and tropospheric column NO$_2$ from FP7 QA4ECV OMI, v1 (Boersma et al., 2017) to verify consistency in our trend estimates.

We note that using 2°×2° area to represent cities does lead to slight overlap over Guangzhou and Shenzhen, Beijing and Tianjin. This does not affect our analyses of emission inventories because we apply geopolitical maps of city boundaries to calculate emissions for each city (see Section 2.2.2). This does have an impact on our analyses of satellite observations because we use all the grids in the 2°×2° area to conduct the spatial regression. However, we do not expect the overlap to significantly change our results because (1) the overlapped area is relatively small; (2) the overlapped cities are sometimes considered together as a whole region because of their similarities and connections (for example, the Jing-Jin-Ji megalopolis and the Pearl River Delta), and (3) the overlapped cities are in the same classes with similar patterns based on our analyses (i.e., Beijing and Tianjin are both in class 2, while Guangzhou and Shenzhen are both in class 4; Table 2).

### 2.2.2 Emission Inventories and Model Simulations

Multiple bottom-up emission inventories for CO, NO$_2$ and SO$_2$ are analyzed, namely Emission Database for Global Atmospheric Research (EDGAR, Crippa et al., 2016), Representative Concentration Pathways (RCP8.5, Riahi et al., 2011), Regional Emission inventory in ASia (REAS) version 2.1 (Kurokawa et al., 2013), and Hemispheric Transport of Air Pollution (HTAP, Janssens-Maenhout et al., 2015). We also use top-down emission estimates of CO and NO$_2$ from the Tropospheric Chemical Reanalysis (TCR) based on CHASER-LETKF assimilation system (Miyazaki et al., 2017). Since these emission inventories have different spatial resolutions (see details in Table 1) and are available in the form of fluxes (units in kg/m$^2$/s), we upscale/downscale them by simply regridding into 0.1° by 0.1° cells similar to our approach for satellite data to facilitate comparison. We then consider all cells within the 2° by 2° area around the city center. For annual emissions, we only take the sum of all cells within the geopolitical boundary of the city (see Figure 2). All of the cities extend to less than the 2° by 2° area that we set as our city domain.

We also use model data for CO and NO$_2$ from the Community Atmosphere Model with Chemistry (CAM-chem; Gaubert et al., 2016) and TCR to derive CO and NO$_2$ abundance ratios associated with the bottom-up emissions used in these models (i.e., RCP in CAM-Chem and EDGAR in CHASER). The associated retrieval averaging kernels and prior information are applied to the daily-averaged model CO and NO$_2$ vertical profiles of mixing ratios from CAM-chem and CHASER, along with appropriate spatial interpolation and/or partial column integrations. Since the spatial resolution (about 2°~3°) of CAM-chem and CHASER outputs that we analyzed are far coarser than 0.1°, we only considered the associated

abundance ratio rather than deriving enhancement ratio across the month where non-stationarity and non-linearity issues are more likely to exist.

## 2.3 Deriving Enhancement Ratios using Spatial Regression Analysis

For each city, we regress the gridded monthly-average CO and $SO_2$ to $NO_2$ to calculate monthly
enhancement ratios ($\Delta CO/\Delta NO_2$ and $\Delta SO_2/\Delta NO_2$). We use $NO_2$ as our control variable as $NO_2$ has the shortest lifetime (hours) among these products. Except for lightning, $NO_X$ is mostly produced from high-temperature anthropogenic combustion processes. And because of its short lifetime, it is observed as distinctly and spatiotemporally local surface enhancements, with relatively very low background concentrations. Along with the availability of $NO_2$ retrievals from satellites at fine spatial scale and over
long period, $NO_2$ allows us to effectively identify intra-megacity combustion activities and define the urban extent (Bechle et al., 2011; Lamsal et al., 2013; Hakkarainen et al., 2016). In other words, $NO_2$ is a good proxy for combustion activity. We use a reduce major axis regression (Smith, 2009) to estimate the slopes ($\Delta y/\Delta x$) representing enhancement ratio across the spatial extent of the megacity, and intercept ($y^{bg}$) for CO and $SO_2$ representing the background levels when there is no combustion (within the
megacity and free-tropospheric contribution). This follows the approach introduced by (Fujita et al. (1992) and Parrish et al. (2002). However, we note that we use the spatial covariations of these species relative to $NO_2$ rather than their temporal covariations as in previous studies. Please see Section 3 for implications of this approach. We only consider statistically significant and positive slopes as we are focusing on sources and not sinks of these combustion products. These monthly ratios are then averaged across the
year for analysis and archived for time series (decadal) analysis (see Section 3). Note that they can be considered to be comparable to emission ratios when observations are taken at or near the source and if they are normalized to account for air mass variations (Fujita et al., 1992; Parrish et al., 2002; Parrish, 2006; Hassler et al., 2016). Here, we normalize all ratios to year 2005 values.

It is important to note that we view each large city as a big smokestack that emits an aggregate of
combustion products that can then be observed by satellite remote sensing as column-integrated quantities. The spatial (0.1˚) covariation of these aggregate within the 2˚ radius is interpreted as bulk characteristic of spatially heterogeneous combustion sources within the megacity. Monthly enhancement ratios are hence interpreted as the linear sensitivity in CO or $SO_2$ to intra-megacity spatial variations in combustion activity as defined by $NO_2$. We emphasize that these enhancement ratios are not derived using time
covariations but spatial covariations to minimize potential non-stationarities (e.g., differences in lifetimes between species), and influence of free-tropospheric signatures in MOPITT CO, which should be reflected as part of a larger scale contribution to $CO^{bg}$ in this analysis given that we anchor the regression on OMI $NO_2$. Possible confounding factors such as biogenic sources of CO in a megacity is also minimized in our analysis by treating CO data only when $NO_2$ is observed since $NO_2$ is not largely co-
emitted from CO biogenic sources. Although spatial and temporal smoothing can minimize the effect of lightning ($NO_X$) and fires ($NO_X$ and CO) since they are emitted intermittently relative to anthropogenic combustion, our findings must be interpreted to represent changes in bulk combustion cleanness over a megacity rather than specific combustion cleanness.

**2.4    Time Series Analysis and Curve Fitting**

The focus of this work is to study the long-term changes in the spatial covariations of these monthly-averaged CO and $SO_2$ to $NO_2$, as expressed in terms of enhancement ratios. We hypothesized that at decadal scale the changes in covariations reflect the dominant changes in megacity emission characteristics. We use two approaches to calculate the decadal trend in our normalized estimates of these ratios. For linear trend analyses, we use the Robust Regression Using Iteratively Reweighted Least-Squares (Holland and Welsch, 1977). This minimizes the influence of outliers relative to traditional least-squares fit especially when the relationship is not fully linear. We also use another trend analysis algorithm in our subsequent inverse analysis. Instead of using the annual mean values and estimate the linear trend across 2005-2014, we estimate the associated decadal trends in $\Delta CO/\Delta NO_2$ and $\Delta SO_2/\Delta NO_2$ using the seasonal trend decomposition with LOESS (locally weighted scatterplot smoothing) or STL algorithm (Cleveland et al., 1990). This algorithm separates the seasonal, inter-annual, and decadal contributions of monthly ratios. We use the smoothing windows for the decadal, inter-annual, and seasonal trends of 121 months, 25 months, and 5 months, respectively based on analysis of CO decadal trends in Jiang et al. (2018). As in Gaubert et al. (2017), we tested several other windows and found consistent temporal patterns across cities. For non-linear curve fitting, we use robust least square regressions with Least Absolute Residuals (LAR) method (within the cftool function in MATLAB) to fit a power law function to the annual-mean ratios and GDP per capita. This method also minimizes the influence of extreme values on the fit.

**2.5    Inverse Analysis**

We conduct an inverse analysis of the long-term trends in monthly enhancement ratios to further expound our findings by associating the overall changes to sectoral changes. In this case, we are interested in finding the decadal contribution of the time series (2005-2014) of monthly statistically-significant enhancement ratios that are derived from our previous regression and time series analysis. We decomposed the *a priori* estimate of monthly emission ratio of CO to $NO_X$ (and $SO_2$ to $NO_X$) from RCP8.5 as a product of: a) ratio of effective emission factors for each of the four sectors (namely energy, industry, transport, and others); and b) fractional contribution of $NO_2$ emissions from each sector to the total $NO_2$ emissions for all four sectors. We then use a two-step Monte-Carlo-based Bayesian inversion method, to estimate effective emission factors and fractional contribution of $NO_2$ emissions from each sector. Please refer to Appendix A for a short derivation of this decomposition, and Appendix B for details in the inverse analysis.

**3    Results and Discussions**

**3.1    Observed Patterns of Enhancement Ratios in Chinese and U.S. Cities**

In this sub-section, we present observed patterns of enhancement ratios in Chinese and U.S. Cities. We firstly show spatial regression analysis of satellite retrievals of CO and $SO_2$ to $NO_2$ by season (taking Beijing and Los Angeles for demonstration) in Figure 2. Although naturally-produced CO and $NO_2$ like

biogenic CO and lightning $NO_X$ introduce a strong seasonality on these ratios even within the megacity, we find that when we average the monthly ratios using only the months corresponding to a particular season (i.e., more fires and lightning during the summer), we still find a similar temporal pattern (albeit different in magnitude) in derived $\Delta CO/\Delta NO_2$ and $\Delta SO_2/\Delta NO_2$ (see Figure 2). This is reasonable as these
CO as well as $SO_2$ enhancements are dominantly from combustion-related processes that co-emit $NO_2$ by our study design, pointing to the robustness of analyzing annual-mean $\Delta CO/\Delta NO_2$ and $\Delta SO_2/\Delta NO_2$.

      Shown in Figure 3 are linear trends of annual-mean $\Delta CO/\Delta NO_2$ and $\Delta SO_2/\Delta NO_2$ relative to year 2005 values in four Chinese cities. These cities are representative of a certain level of urban development across mainland China. The four levels in this study are defined using broad clustering between the
average GDP per capita per year and the rate of change in $\Delta CO/\Delta NO_2$ that are derived from satellite observations. This is shown in Table 2, where a general rule resulting from this analysis would be a classification mainly based on GDP per capita per year, except Harbin and Wuhan. Combustion-related activities in Shenyang, Beijing, Shanghai, and Shenzhen can be characterized to follow a progression from heavy to light manufacturing, export processing, and service industries (Chan and Yao, 2008). For
this analysis, Shenyang, Beijing, Shanghai, and Shenzhen represent the progression across the 12 select cities of increasing GDP per capita along with decreasing to increasing $\Delta CO/\Delta NO_2$ (-5.4±0.7%/year to +8.3±3.1%/year) and decreasing rate of $\Delta SO_2/\Delta NO_2$ reductions (-6.0±1.0%/year to -3.4±1.0%/year) relative to 2005 (Figure 3 and Table 2). This pattern in enhancement ratios is not evident in the rate of change of CO, $SO_2$, and $NO_2$ column abundance, for which we find increasing rate of decrease in CO (-
0.1±0.3%/year to -1.0±0.2%/year) and $SO_2$ (-1.9±0.9%/year to -5.5±1.1%/year) abundance, along with decreasing rate of increase in $NO_2$ abundance from Shenyang (+5.2±1.4%/year) to Shenzhen (1.8±0.7%/year) (Table 2). This is consistent with previous studies of these species. In fact, we find a decreasing-to-increasing pattern in the derived enhancements of CO due to combustion (i.e., $\Delta CO_{comb} = \langle CO - CO^{bg}\rangle$), across these four levels of development.
25       We have minimized the influence of inter-annual variations due to meteorology (e.g., changes in air mass) by analyzing molar ratios (e.g., mole CO/mole $NO_2$) rather than absolute molar concentrations (e.g., mole CO/mole air; Parrish et al., 2002, 2006). As the co-emitted species (i.e., CO, $SO_2$, and $NO_2$) are subject to the same meteorological conditions (affecting transport, dilution, and lifetime), their enhancement ratios are expected to be less sensitive to meteorology compared to the absolute molar
concentrations. This is supported by the fact that decadal $\Delta CO/\Delta NO_2$ as well as $\Delta SO_2/\Delta NO_2$ for different seasons have similar trends (Figure 2). Previous studies have also proven that the ratios compared to the concentrations themselves are relatively immune to changing meteorological conditions, and can provide insights into the magnitude and temporal trends of the emissions (Parrish et al, 2002, 2006, 2009, Silva et al., 2013, Hassler et al. 2016). In addition, they can be directly compared to the corresponding emission
ratios under certain circumstances. However, we note that even though the ratios derived from satellite observations are relatively less sensitive to meteorology, the methodology cannot eliminate all the impacts from meteorology. The enhancement ratios may be impacted by the meteorological conditions because lifetimes of different air pollutants may respond to meteorological conditions differently. Nevertheless, we believe such impact should not influence our main conclusions for the following two reasons: (1) Our
analysis focuses on decadal trends instead of short-term trends. As shown by previous study, meteorology also plays an important role on relatively short time scales, but meteorology probably plays a lesser role in the longer-term trends (Krotkov et al. 2016); (2) The satellite retrieval samples are taken over the

megacities (right above strong emission sources) instead of downwind of the pollution sources, making them more representative of megacity sources.

Normalizing these ratios to 2005 values should have also minimized the impact of the differences in the magnitude of these ratios between these cities. The impact of meteorology on inferred decadal trends through variations in columnar abundance is more evident when absolute magnitudes of single species are analyzed. In addition, potential drifts of biases in time (caused by systematic errors in the instrument and/or retrieval algorithm) cannot account for the differences in the temporal pattern that we find across these cities. Such biases should be commonly reflected in all cities, yet we see differences between cities. In fact, we find very similar progression pattern when we use the Infrared Atmospheric Sounding Interferometer (IASI) CO retrievals (De Wachter et al., 2012) instead of MOPITT, or OMI QA4ECV (Boersma et al., 2017) instead of OMI DOMINO. Interestingly, we find that the increasing enhancement ratio of CO to $NO_2$ in Shenzhen (and to a lesser extent in Shanghai) remarkably resembles the relative changes in CO to $NO_2$ ratios in more developed megacities (Los Angeles and New York) and several urban agglomerations in the United States (see Figure 3e for Los Angeles and Table 2 and Figure S1 for all other select cities). More importantly, the increasing pattern that we see in Los Angeles ($\sim$ +7$\pm$1%/year) relative to 2005 is generally consistent to the increasing trend ($\sim$ +4%/year) after 2007 of ground-based CO to $NO_X$ enhancement ratio in Los Angeles as reported by Hassler et al. (2016). It is a common understanding that modernization brings about larger energy use coupled with higher economic productivity, but poorer environmental quality (i.e., increasing abundance of pollutants). However, the changes in lifestyle concomitant with human development results in a shift to fewer activities (including increase use of renewable energy), along with more efficient and cleaner combustion and changes in fuel types (coal to natural gas) (Mazur and Rosa, 1974). This eventually leads to increases in relative sensitivities of CO and $SO_2$ to $NO_2$. Along the same line as previous studies suggesting emissions of CO, $SO_2$, $NO_2$, and their ratios can be indicators of modernization to some extent (Krotkov et al., 2006; Russell et al., 2012; Luo et al., 2014; Hassler et al., 2016), our finding on this progression in $\Delta CO/\Delta NO_2$ serves as a satellite-based evidence of a dominant shift in the cleanness of bulk combustion in more economically developed city within a developing country like China.

On the other hand, there is no clear difference in the observed enhancement ratios ($\Delta SO_2/\Delta NO_2$) and derived enhancements of $SO_2$ due to combustion ($\Delta SO_{2_{comb}}$) between cities. The sensitivity of $SO_2$ to $NO_2$ relative to 2005 in Shenzhen does not follow the increasing pattern in Los Angeles (Figure 3b). Unlike $\Delta CO_{comb}$, $\Delta SO_{2_{comb}}$ in all four Chinese cities still show a decreasing trend relative to 2005 while $\Delta SO_{2_{comb}}$ in Los Angeles show an increasing pattern consistent with its $\Delta CO_{comb}$. On one hand, there is a striking difference in absolute magnitudes in $SO_2$ abundance between these cities (as has been reported), reflecting large-scale differences in combustion practice. Yet, the low $SO_2$ abundance in Los Angeles makes it also difficult to detect possibly large $SO_2$ point sources (Krotkov et al., 2016). Enhanced $SO_2$ signal can still be detected as the spatial first-order derivatives of $SO_2$ with $NO_2$ at megacity-scale should not be largely (non-linearly) influenced by its absolute magnitude. We find that there is a tighter correspondence between $SO_2$ and $NO_2$ abundance in Chinese cities than in U.S. cities. This might suggest differences in fuel use as $SO_2$ is mainly produced within a megacity from burning of sulfur-containing fossil fuel (mostly coal, oil, and natural gas) and to a smaller extent from industrial processes (e.g., smelting). Here, we postulate that the absence of an apparent shift in $\Delta SO_2/\Delta NO_2$ across the four Chinese

cities is due to continuing heavier reliance of these cities (and China) on coal burning relative to United States (Wang and Hao, 2012; Bhattacharya et al., 2015; Qi et al., 2016; Yang et al., 2016; Sun et al., 2018; Zheng et al., 2018). In terms of the sectoral share, the majority of NOx emissions over Los Angeles basin is from transport according to a recent fuel-based inventory (Hassler et al., 2016), whereas fossil fuel combustion (from power generation and industry) is the most dominant NOx source in China (Sun et al., 2018). In terms of the energy share, it was estimated that coal accounts for about 69% and 23% of the total primary energy consumption in China and U.S. in 2005, respectively. Actions including usage of low-sulfur coals, installation of flue gas desulfurization (FGD) facilities, and closing of small units, have been taken to reduce coal-related emissions in China. The aforementioned de-$SO_2$ procedure in China is most likely to be the dominant driving factor of the declining $\Delta SO_2/\Delta NO_2$ (Li et al., 2018; Zheng et al., 2018). While there are on-going activities regulating coal-related emissions, coal consumption in China remains to increase in the past decade (Qi et al., 2016; Yang et al., 2016). In terms of mass, it has increased by 70% from 2005 to 2014 (Korsbakken et al., 2016). On the other hand, the use of coal in U.S. has been found to be slightly decreasing along with previous adoption of $SO_2$ control technologies (Taylor et al., 2005). In addition, previous studies have reported recent reduction in NOx emissions over China since 2011 based on satellite observations and emission inventories (Liu et al., 2016; van der A., et al, 2017). The installation of selective catalytic reduction (SCR) equipment at power plants and new emissions standards for vehicles both contribute to the NOx emission reduction (Liu et al., 2016; van der A., et al, 2017; Wu et al., 2017). On the other hand, based on our analysis of decadal trends (2005-2014), only $NO_2$ over Shenzhen overall decreased in the decade, while 10-year average changes of $NO_2$ over Shenyang, Beijing, and Shanghai were overall positive (Table 2). Intradecadal changes as reported in Liu et al. 2016 (from increasing to decreasing NOx emissions around 2011) do not contradict the derived 10-year trend in this work, especially over Shenyang, and Beijing where NOx emissions are still rapidly increasing during the first half of the decade (2005-2011). The changes in $SO_2$ emissions and $NO_2$ emissions together contribute to the trends of $\Delta SO_2/\Delta NO_2$ that we found. Positive $\Delta NO_2$ and negative $\Delta SO_2$ produce negative $\Delta SO_2/\Delta NO_2$ over the three cities; while negative $\Delta SO_2$ and negative $\Delta NO_2$ (albeit smaller in magnitude) still produce negative $\Delta SO_2/\Delta NO_2$ but smaller magnitude over Shenzhen than $\Delta SO_2/\Delta NO_2$ over the other cities (Table 2). This indicates a stronger influence of the changes in $SO_2$ emissions (as reflected in $\Delta SO_2$) in the decreasing trends of these ratios.

## 3.2    Inconsistencies with A Priori Estimates

The satellite-based $\Delta CO/\Delta NO_2$ patterns are inconsistent with emission- and model-based ratios (Figures 3 and S1, Table 2). As previously introduced, estimates of the ratios of emissions can be related to observed ratios of enhancements when these observations are taken at or near the source. In this case, we assume that a megacity is a big smokestack emitting mostly combustion-related pollutants (i.e., CO, $NO_2$, and $SO_2$) that can be observed from space with MOPITT and OMI. In addition, $NO_2$ is considered to be the dominant form of $NO_X$ that can be observed at this scale. From a global atmospheric chemistry modeling (CTMs) perspective, the associated abundance over megacities is represented as one to four discrete vertical column(s) assuming spatial resolution of these CTMs of one to two degrees. While recognizing the associated month-to-month variability in $\Delta CO/\Delta NO_2$ and expected differences on how these ratios should be compared, the trends in emission ratios relative to 2005 of CO to $NO_X$ from bottom-

up emission inventories (EDGAR4.2 and RCP8.5) and top-down emission estimates (CHASER, Miyazaki et al., 2017) do not appear to follow the progression (i.e., decreasing to increasing $\Delta CO/\Delta NO_2$ relative to 2005 from Shenyang to Shenzhen; Figure 3). This is also true for the ratios of CO to $NO_2$ abundance from CAM-Chem and CHASER CTMs, which are mostly consistent (except in Los Angeles)
with the trends of their associated emission ratios (i.e., CAM-Chem and CHASER emissions are based on RCP8.5 and EDGARv4.2 inventories, respectively). The *a posteriori* emission ratios in Beijing from Miyazaki et al. (2017), which uses CHASER-LETKF to assimilate MOPITT CO and OMI $NO_2$ retrievals among other retrievals, also appear to initially follow the emission ratios from EDGAR. Furthermore, the ratios of $SO_2$ to $NO_X$ emissions from RCP8.5 follow the trend of $\Delta SO_2/\Delta NO_2$ in Chinese cities but tend
to diverge in Los Angeles, whereas the emission ratios from EDGAR exhibit a lack of trend in China and Los Angeles. A closer look at linear trends of the ratios for each sector in RCP8.5 (Figure S2) reveals inconsistencies in the trends, which cannot be addressed by simple scaling of activity levels in bottom-up inventories (Zheng et al., 2018). All these differences underscore the need to reduce uncertainties in representing time-varying emission activity and emission factors in CTM inputs. There is also a need to
quantify errors in model physics and dynamics in transforming emissions to abundance, as well as in data assimilation and inverse methods in integrating observations into models including representativeness of these retrievals. We highlight here the need to improve not only the accuracy but also the consistency of AQ predictions across pollutants in megacities. Initial results from an improved set of multi-species data assimilation runs using CHASER-LETKF show better agreements with the trends in $\Delta CO/\Delta NO_2$
(Miyazaki et al., 2017). Such improvements highlight an under-explored utility of available observational constraints on the changes in emission ratios. We emphasize here that while these differences are expected and have been previously reported, our findings highlight the need to focus on improving model treatments of the dynamic nature of emission factors in these megacities.

### 3.3    Combustion Emission Pathway for Chinese Cities

25       We define combustion emission pathway as a trajectory in time of the overall changes in emissions due to combustion with respect to socioeconomic development (e.g., Riahi et al., 2011; Steinberger et al., 2012; Li et al., 2016; Marangoni et al., 2017). In this section, we identify a common combustion emission pathway across these four levels of development and associate them to sectoral changes through inverse analysis. We will briefly describe the inverse analysis of the ratios in section 3.3.1, present our
findings on combustion emission pathway in section 3.3.2, and elucidate the driving factors by means of time traces in sectoral emission ratios in section 3.3.3.

### 3.3.1    Inverse Analysis of the Ratios

      We conduct an inverse analysis of the ratios shown in Figure 3 to further expound on these patterns, by associating them to sectoral changes. Please see details of the matrix-vector product and inversion
methodology in Section 2.5 and Appendix B. The result of this inversion is a set of *a posteriori* time series estimates of sectoral CO to $NO_X$ and $SO_2$ to $NO_X$ ratios, such that the corresponding time series estimates of the total CO to $NO_X$ and $SO_2$ to $NO_X$ ratios match the decadal trends of $\mathbf{\Delta ra/\Delta ra_2}$ and $\mathbf{\Delta an_2/\Delta an_2}$ inferred from these satellite retrievals. Again, we note that we use the STL-inferred decadal

trend as the data to fit (not the monthly-mean ratios nor the linear trend in Figure 3), as this is the most appropriate data for analyzing long-term changes in emission sectors.

### 3.3.2 Combustion Emission Pathway

The results of our inverse analysis are presented in Figure 4. This figure consists of five 2-D line
plots of *a posteriori* (solid) and *a priori* (dashed) time series of $SO_2$ to $NO_X$ emission ratios ($ESO_2/ENO_X$) in y-axis versus corresponding values of CO to $NO_X$ emission ratios ($ECO/ENO_X$) in x-axis. The five plots correspond to the annual total (center panel, Figure 4a) and sectoral emission ratios (four side panels, Figures 4b to 4e) of each of the four cities selected in Figure 3. The time series, which is normalized to 2005 values, starts at the origin (1,1) and ends at the arrow tip of the line. Each 2-D plot also contains an
inset showing the corresponding emission trajectory for Los Angeles. The center panel of Figure 4 is similar to Figure 3 but plotted jointly and with the *a posteriori* time series of emission ratios now corresponding to the time series of enhancement ratios (i.e., STL-inferred decadal trend). We find that the progression in combustion characteristics across these four cities is clearly evident from this diagram and very consistent with the linear trends in Figure 3. In Shenyang, both $ESO_2/ENO_X$ and $ECO/ENO_X$
are decreasing relative to 2005 at a faster rate (as represented by the length of the line) than in Beijing. On the other hand, we see a clear shift in Shanghai and most notably in Shenzhen to a slightly decreasing $ESO_2/ENO_X$ and increasing $ECO/ENO_X$ leading their emission trajectories toward a different state of 'combustion cleanness'. The combustion emission ratios in Los Angeles (and other cities in U.S.) lies however at a different state than Shanghai and Shenzhen. In particular, we find $ESO_2/ENO_X$ and
$ECO/ENO_X$ in Los Angeles to be both linearly increasing relative to 2005 values. And so, there exists a progression of decreasing-to-increasing sensitivities of CO and $SO_2$ to $NO_2$ from Shenyang to Shenzhen to Los Angeles (gray semi-circular trace in Figure 4a) relative to 2005, that appears to be related to socioeconomic development consistent with the current understanding of human development pathways (Lamb et al., 2014). In this case, it may be a consequence of air quality management practice and improved
efficiency in China (Sun et al., 2018; van der A et al., 2017) and U.S. (Hassler et al., 2016; Russell et al., 2012). Altogether, this leads us to suggest a common combustion emission pathway for the megacities in mainland China, that begins with a reduction in $SO_2$, followed by CO, and continues with a reduction in $NO_X$ and potentially on volatile organic compounds (VOCs) later on. To illustrate, we still see increases in $NO_X$ abundance in Shenyang although CO and $SO_2$ are already decreasing, whereas in Shenzhen, we
see $NO_X$ starting to decrease (at a faster rate) along with decreasing CO and $SO_2$ abundance. The rate at which $SO_2$, CO, and $NO_2$ are decreasing is not at a level that is observed in Los Angeles. And so, while the satellite data reveals a combustion emission pathway in these Chinese megacities, these cities are yet to reach conditions that is at par with megacities in more developed cities in U.S. and Europe. It is worth noting that the *a priori* estimates from RCP8.5 do not follow this pathway, even for Los Angeles,
suggesting inconsistencies and necessary updates on temporal changes in emission factors, effectiveness in pollution control technologies, and/or more information on fuel use mixtures in this emission inventory. It also appears that the pathway represented in RCP is similar to all cities and more resembling the emission pathway for Beijing.

### 3.3.2   Traces in Sectoral Emission Ratios

Furthermore, the traces in sectoral emission ratios from RCP8.5 all point to decreasing ratios relative to 2005 and are primarily driven by the energy (transportation) sector, which constitute more than one-third of $NO_X$ emissions in Chinese (U.S.) cities (Figure 4b to 4e). Our inversion results to slight
adjustments in Chinese energy emission pathway towards little to no changes in CO to $NO_X$ emission ratios (Figure 4b). Adjustments from the transportation sector are also small in terms of direction and slower in terms of its rate of change relative to 2005 RCP values (Figure 4c). This is certainly not the case in Los Angeles where CO to $NO_X$ and $SO_2$ to $NO_X$ ratios follow quite the opposite pathway of increasing ratios from the energy sector and increasing CO to $NO_X$, with no change in $SO_2$ to $NO_X$ from the
transportation sector. This is expected in United States because of cleaner fuel standards (Shindell et al., 2011; Zhang et al., 2012; Kheirbek et al., 2014; Yang et al., 2016; Paulot et al., 2017). Significant shifts on these ratios relative to 2005 are clearly evident from the industry and other (i.e., agriculture, residential, and waste) sectors in the cities in China (Figure 4d and 4e). Shanghai and most notably Shenzhen show a shift to increasing CO to $NO_X$ with slightly decreasing $SO_2$ to $NO_X$ that are not reflected in RCP8.5.
The emission ratios from industry and other (mostly residential) sectors need to be adjusted significantly in our inversion to match the shifts in observed $\Delta CO/\Delta NO_2$ and $\Delta SO_2/\Delta NO_2$ in these two cities. As earlier mentioned, tertiary (service) industries including export processing activities are dominant in Shanghai and Shenzhen than in Shenyang. The shift in recent years to increasing CO to $NO_X$ reflects a larger rate of decrease in $NO_X$ levels than CO from the industrial and residential sectors of these cities.
While a more detailed investigation is warranted to narrowly identify the activities and/or policies driving this shift (van der A et al., 2017), it is clear that changes in combustion activity alone cannot account for these shifts, and that updates on emission factors for these sectors in RCP8.5 are needed. We find that these findings are robust across a suite of error assumptions in the inverse analysis. This update applies all the more to all sectors in RCP emissions for Los Angeles. Again, this is well supported by studies like
Hassler et al. (2016). where they reported increasing CO to $NO_X$ enhancement ratio after 2007 in Los Angeles along with a 45% decline of $NO_X$ emissions based on their fuel-based inventory. This is in contrast to decreasing RCP8.5-based MACCity emission ratios that they also reported for Los Angeles. This increase in enhancement ratios (similar to this work) is attributed to a combination of factors such as the decrease in $NO_X$ from freight traffic activity during U.S. recession and implementation of new $NO_X$
emission control technologies and regulations to meet Tier two emission standards on U.S. light-duty vehicles. They also noted that differences in the trends of $\Delta CO/\Delta NO_2$ are still observed even between cities from developed countries like U.S. and Europe, as these cities differ in terms of transportation practices and lifestyles (e.g., increase in light duty diesel vehicles). It is also now conceivable that $\Delta CO/\Delta NO_2$ can be further influenced by shifts in relative importance of emission sectors (e.g., VOCs in
petrochemical and pharmaceutical industries) as activity decreases with efficiency, pollution is controlled, and lifestyle changes whenever cities evolve (McDonald et al., 2018). A recent study (Jiang et al., 2018) revealing an over-estimation in the decrease of USEPA $NO_X$ emissions based on OMI $NO_2$ and MOPITT CO retrievals with USEPA ground station measurements of $NO_2$, also suggests potential changes in 'bulk' combustion characteristics in urban regions of the United States. Along with these studies, our results
suggest that regional to global emission inventories, which are used as input to predictive models of atmospheric composition, have to reflect: a) the evolution of air pollution for a given city (sectoral shifts)

and b) the differences in combustion practices from city to city, in order to capture these observed magnitudes and variations in enhancement ratios.

### 3.4 Socioeconomic Dependence of Urban Enhancement Ratios in China

Here, we attempt to connect these emission pathways to the larger pattern of economic growth across the 31 capital cities and five special cities in mainland China. We find in particular a power law relationship between the observed annual-mean $\Delta CO/\Delta NO_2$ (and $\Delta SO_2/\Delta NO_2$) and GDP per capita. This is not to derive an overall EKC for China, as this in fact requires a very long record of environmental quality, but specifically to investigate how economic development shapes how 'clean' the bulk combustion in Chinese cities would be. These enhancement ratios complement abundance and/or emissions of pollutants as traditional measures of air pollution. Unlike Figure 3 and 4, our focus is to illustrate the larger dependence of enhancement ratios on GDP per capita. As discussed in the Methods section, we relate the enhancement ratio of a megacity to the ratio of the product of emission factor ($EF_{species}$) and effectiveness of control technology ($1 - CE_{species}$) for CO and $NO_X$ species in the case of $\Delta CO/\Delta NO_2$ for example. We use a robust least-squares regression with least absolute residuals method to fit a curve of the form: $y = ax^k$, where $y$ is $\Delta CO/\Delta NO_2$ or $\Delta SO_2/\Delta NO_2$ and $x$ is GDP per capita. Our results are presented in Figure 5a and 5b for $\Delta CO/\Delta NO_2$ and $\Delta SO_2/\Delta NO_2$, respectively. The 12 cities considered in our analysis of emission pathways are marked with colors corresponding to its level of urban development described in previous section. Note that the magnitudes of enhancement ratios derived from this work is a factor of 10 higher than ratios derived from ground-based networks. We attribute this discrepancy to differences in air mass and volume, representativeness, and vertical sensitivity between abundance retrieved as total or tropospheric columns and in-situ and point samples in units of mixing ratios. Nevertheless, we find a strong power law relationship with GDP per capita having $k$ coefficients ($R^2$=0.98) of negative two-thirds and negative one-half for $\Delta CO/\Delta NO_2$ and $\Delta SO_2/\Delta NO_2$, respectively. Likely, the coefficients in $\Delta SO_2/\Delta NO_2$ will converge to that in $\Delta CO/\Delta NO_2$ as changes in fuel type and $SO_2$ controls should decrease $SO_2$ abundance. While each city is unique and that the evolution of air pollution may be different from city to city, there also exist a clear signature of urbanization at national level that reflects the influence of economic growth on the cleanness of bulk combustion. Similar power law relationships (albeit different coefficients) have been reported in studies of urban growth and development (Bechle et al., 2011; Lamsal et al., 2013; Bettencourt et al., 2013), energy flows (Creutzig et al., 2015) and carbon emissions (Fragkias et al., 2013). Our results suggest that enhancement ratios scale with GDP per capita, with lower GDP per capita like Shenyang and other cities (gray dots) having higher enhancement ratios, while Shenzhen and other cities (yellow dots) with highest GDP per capita in China lie among cities with the lowest enhancement ratios. As we have shown in Figure 4 (and Table 2), the ratios in Shenzhen tend to increase with time (and GDP) but this increase has its limits and appears to be dwarfed by cities with highest enhancement ratios. We note, however, that identifying a mechanistic rationale of these negative scaling coefficients is beyond the scope of this work and hence is not proposed. A unified relationship cannot also be established across countries as there are obvious differences in socioeconomic and air pollution conditions in China and U.S. that cannot be accounted for (Figure 6). Nevertheless, we suggest incorporating this observable along with estimates of emissions to future scaling studies, especially as we move past RCPs and toward recent developments in building more realistic

emission scenarios that integrate socioeconomic and environmental development pathways like the Shared Socioeconomic Pathways (SSPs; O'Neil et al., 2014).

## 4     Summary and Implications

The main goal of this work is to provide observational evidence from Earth observing satellites of emission pathways of combustion-related air pollutants, as a result of urban growth in economically developing countries like China. A new observational perspective on monitoring one of the major consequences of urbanization is introduced, not to replace existing observing capabilities but to further exploit the information that is already available. Following the pioneering work by Parrish et al. (2002), the sensitivities of intermediate products of combustion can be derived from existing satellite retrievals of air quality (AQ), to inform changes in bulk combustion characteristics (and consequently emissions) of a megacity. This is especially relevant as the number of megacities continue to grow in the coming decades, mostly at locations that lack sufficient AQ monitoring capabilities. Enhancement ratios of CO to $NO_2$ and $SO_2$ to $NO_2$ over megacities in mainland China that are derived from MOPITT and OMI satellite instruments show a coherent long-term progression in recent years of decreasing to increasing ratios relative to 2005. This is well correlated with economic development. These trace a common emission pathway that resembles the evolution of air pollution in more developed cities in the United States which is characterized by transitions in energy use and subsequent implementation of pollution control and regulation. Although we find cleaner combustion as cities in China develop consistent with their Five Year Plans, this is presently obfuscated by increasing fuel use particularly its heavy reliance on coal. We propose the use of these enhancement ratios derived from existing satellite retrievals to complement existing surface AQ networks, including carbon-related satellite observing systems in further constraining combustion efficiency and effectiveness of control technologies and policies. Augmenting existing capabilities (Saeki et al., 2017) is particularly relevant, especially with the aid of big data informatics and machine learning as well as the advent of activities focusing specifically on tracking fossil fuel emissions (like the $CO_2$ Human Emissions project; https://www.che-project.eu). While we recognize the current limitations of these retrievals (e.g., collocation, sensitivity), our findings appear to be robust across retrievals and methods, and are supported by previous studies using these retrievals in a different way (Krotkov et al., 2016; Jiang et al., 2018) or ground measurements (Hassler et al., 2016). We strongly suggest that the capability to monitor relatively long-term changes in atmospheric composition has to be supported and continued with complementary new satellite and field missions and deployments (Streets et al., 2013; National Academies of Sciences, Engineering, and Medicine, 2016).

The relative importance of monitoring combustion efficiency and effectiveness of pollution control increases as a city and country continue to socioeconomically develop and become sustainable. Despite past and present studies (Mazur and Rosa, 1974; Lamb et al., 2014), it is only in most recent years that we have developed comprehensive and integrated monitoring and prediction systems, which paved new measures of air pollution and new developments in emission scenarios like SSPs. For China, more detailed information on energy use and improved emission inventories are increasingly becoming available for assessment (Li et al., 2017; Zhong et al., 2017). As we also recognize some of the challenges to quantify socioeconomic variables such as the impact of international trade on air pollution (Lin et al.,

2014), economic structural upgrading (Mi et al., 2017), greater utilization of renewable energy, and even metrics of performance (Ramaswami et al., 2013), from a physical science perspective, our results strongly support these new developments. We find inconsistencies between the long-term spatiotemporal patterns of emission ratios from RCP8.5 and model predictions of abundance ratios, and the corresponding patterns derived from observed enhancement ratios. Scientific improvements in representing the evolution of air pollution (Lewis, 2018) and emission pathways (Mitchell et al., 2017) can be made by (1) considering observationally-constrained time-varying emission factors, and (2) confronting emissions and physical models with available data not only for their accuracy, but also for their consistency in representing both carbon and AQ-related combustion products.

*Data availability.* The raw data used in this study are available online (links to satellite data and emission inventories can be found in Table 1; Socioeconomic Data: Annual GDP and population are directly taken from China Statistical Yearbook compiled by National Bureau of Statistics of China (http://www.stats.gov.cn/)). Model outputs and reanalysis data are available upon request from the authors.

*Acknowledgments.* This study is supported by NASA ACMAP Grant NNX17AG39G. K.M's reanalysis is supported by JSPS KAKENHI Grant 15K05296 and 18H01285. We acknowledge MOPITT, IASI, and OMI retrieval teams for CO, $NO_2$, $SO_2$, data, respectively. We also thank EDGAR, HTAP, REAS, RCPs data teams for the emission inventories. All the satellite data and emission inventories are available to the public online. We thank Kevin Bowman, Cenlin He, and Sam Silva for insightful discussions.

*Author contributions.* The initial idea was provided by AFA. and WT. CHASER-LETKF experiments were performed and provided by KM. CAM-chem modeling experiments were performed and provided by BG. Data analyses and results interpretation were performed by WT. and AFA. HW provided key expert guidance on MOPITT CO. The manuscript was written by AFA. and WT.

**Appendix A. Combustion Emission Ratios and their Decomposition**

In a combustion process using a hydrocarbon fuel, CO and elemental carbon (e.g., soot or BC) are produced when combustion is incomplete; otherwise carbon in the fuel is oxidized to $CO_2$ (Eq. 1). In addition, NO and $NO_2$ are produced from the oxidation of nitrogen from the fuel itself and from decomposition of $N_2$ in air at high temperatures (Flagan and Seinfeld, 2012). Sulfur dioxide ($SO_2$) is also produced when the fuel used in the combustion process contains sulfur (such is the case for low-grade fuels).

$$C_{x_1}H_{x_2}O_{x_3}N_{x_4}S_{x_5} + n_1(1+e)(O_2 + 3.76N_2) \rightarrow$$
$$n_2CO_2 + n_3H_2O + n_4O_2 + n_5N_2 + n_6CO + n_7NO + n_8NO_2 + n_9SO_2 + n_{10}C + \cdots \qquad \text{Eq. (A1)}$$

Emissions of these intermediate product are typically expressed as:

$$E_x = \sum_s \left[ A_s \cdot EF_{x,s} \cdot \left(1 - CE_{x,s}\right) \right]$$

$$= \sum_s \left[ A_s \cdot EEF_{x,s} \right] \qquad \text{Eq. (A2)}$$

where $E_x$ is the total mass of emissions for species $x$, $EF_{x,s}$ is its associated emission factor for a specific source/sector $s$, $A_s$ is the activity level of the source. $CE_{x,s}$ corresponds to effectiveness of control measure and $EEF_{x,s} = EF_{x,s} \cdot \left(1 - CE_{x,s}\right)$ is the effective emission factor. When we take the ratio of emissions (Eq. 2) of co-emitted species $x$ and $y$,

$$\frac{E_y}{E_x} = \frac{\sum_s \left[ A_s \cdot EEF_{y,s} \right]}{\sum_s \left[ A_s \cdot EEF_{x,s} \right]} = \sum_s \left( \frac{EEF_{y,s}}{EEF_{x,s}} \right) \left( \frac{E_{x,s}}{E_{x,total}} \right) \qquad \text{Eq. (A3)}$$

this ratio can be expressed as the sum of the products of the ratio of effective emission factors ($R_{x,y,s}^{EEF}$) and the fractional contribution of emission sector f for species x ($f_{x,s}$) (Eq. A3).

## Appendix B. Inverse Analysis

We decomposed the *a priori* estimate of monthly emission ratio of CO to $NO_X$ (and $SO_2$ to $NO_X$) from RCP8.5 as a product of: a) ratio of effective emission factors for each of the four sectors namely energy, industry, transport, and others ($R_{x,y,s}^{EEF}$); and b) fractional contribution of $NO_2$ emissions from each sector to the total $NO_2$ emissions ($f_{x,s}$) for all four sectors $s$ ($s_1$: energy, $s_2$:industry, $s_3$:transport, $s_4$:others). In matrix-vector form, this can be expressed as:

$$\begin{bmatrix} ECO/ENO_X \\ ESO_2/ENO_X \end{bmatrix} = \begin{bmatrix} R_{CO/NO_X,S_1}^{EEF} & R_{CO/NO_X,S_2}^{EEF} & R_{CO/NO_X,S_3}^{EEF} & R_{CO/NO_X,S_4}^{EEF} \\ R_{SO_2/NO_X,S_1}^{EEF} & R_{SO_2/NO_X,S_2}^{EEF} & R_{SO_2/NO_X,S_3}^{EEF} & R_{SO_2/NO_X,S_4}^{EEF} \end{bmatrix} \begin{bmatrix} f_{NO_X,S_1} \\ f_{NO_X,S_2} \\ f_{NO_X,S_3} \\ f_{NO_X,S_4} \end{bmatrix} \qquad \text{Eq. (B1)}$$

or

$$\mathbf{y} = \mathbf{Hx} \qquad \text{Eq. (B2)}$$

We use a two-step Monte-Carlo-based Bayesian inversion method to estimate both $\mathbf{H}$ and $\mathbf{x}$ of the following cities: Shenyang, Beijing, Shanghai, Shenzhen, and Los Angeles. We focus our analysis on the decadal trends of the RCP8.5 CO to $NO_X$ and $SO_2$ to $NO_X$ emission ratios using the decadal trends of $\Delta CO/\Delta NO_2$ and $\Delta SO_2/\Delta NO_2$ as observational data ($\mathbf{y}$). We use the decadal trend of enhancement ratios of CO to $NO_2$ and $SO_2$ to $NO_2$ (derived using STL), calculate their annual averages and normalized to 2005 values, and then take these as our observational (fitting) data. Our goal is to estimate $\mathbf{H}$ and $\mathbf{x}$ given $\mathbf{y}$ subject to the following constraints: a) errors in $\mathbf{H}$ and $\mathbf{x}$ are 10% and 25% of their values, b) errors in $\mathbf{y}$ is 5% of its value, c) error covariances of $\mathbf{y}$ and $\mathbf{x}$ are uncorrelated and diagonal ($\mathbf{S_e}$, $\mathbf{S_a}$) and d) sum of $\mathbf{x}$ is unity. Since this is an under-determined inverse problem, we apply prior information on $\mathbf{H}$ and $\mathbf{x}$

using the RCP emissions ($\mathbf{H_a}$, $\mathbf{x_a}$). We conduct our inverse analysis into two-step: 1) estimate the most likely $\mathbf{H}$ that results to estimates of $\mathbf{x}$ best fitting the decadal trend, 2) estimate $\mathbf{x}$ using the new estimate of $\mathbf{H}$. For Step 1, first, we draw n=10,000 samples of $\mathbf{H}$ assuming its errors are normally distributed with mean to be its prior and covariance to be the diagonal of its squared errors. Second, we use the *maximum*
*a posteriori* (MAP) solution to the Bayesian problem to estimate $\mathbf{x}$ for every sample. i.e.,

$$\hat{\mathbf{x}} = \mathbf{x_a} + \left(\mathbf{H}_a{}^\mathsf{T}\mathbf{S_e^{-1}}\mathbf{H_a} + \mathbf{S_a^{-1}}\right)^{-1}\mathbf{H_a}{}^\mathsf{T}\mathbf{S_e^{-1}}(\mathbf{y} - \mathbf{H_a}\mathbf{x_a}), \quad \hat{\mathbf{S}} = \left(\mathbf{H_a}{}^\mathsf{T}\mathbf{S_e^{-1}}\mathbf{H_a} + \mathbf{S_a^{-1}}\right)^{-1} \qquad \text{Eq. (B3)}$$

We draw a new sample if any of the elements in $\hat{\mathbf{x}}$ is negative. Third, we take the mean of 100 $\mathbf{H}$ samples resulting to the lowest root-mean-square errors relative to the data. We use this mean as our new
estimate of $\mathbf{H}$ ($\hat{\mathbf{H}}$). For Step 2, we apply the same MAP solution using $\mathbf{x_a}$ and $\mathbf{H_a} = \hat{\mathbf{H}}$ to estimate $\hat{\mathbf{x}}$ and $\hat{\mathbf{S}}$. Similar to a Kalman filter, we cycle this procedure for each year starting from 2006 to 2014. We use the new estimates of $\hat{\mathbf{x}}$, $\hat{\mathbf{H}}$, and $\hat{\mathbf{S}}$ for a given year as priors for the succeeding cycle with fix inflation on the covariance of 1.25 to minimize filter divergence. We note that the additional constraints (positive $\hat{\mathbf{x}}$, sum of $\hat{\mathbf{x}}$ is unity) minimizes the underdeterminacy of the problem. This is supported by post-inverse
analysis diagnostics (i.e., averaging kernels) showing that elements of $\hat{\mathbf{x}}$ are resolved by the trend data. Since $\mathbf{H}$ is drawn based on Monte-Carlo sampling, we do not have a diagnostic for the relative contributions of the prior and the data on $\mathbf{H}$. We chose the mean across 100 $\mathbf{H}$ values resulting to estimates of $\mathbf{H}\hat{\mathbf{x}}$ with the lowest RMSEs relative to the data. The changes in $\hat{\mathbf{H}}$ relative to the $\mathbf{H_a}$ can be explored in the sectoral changes shown in Figure 4. This is especially the case for Shanghai and Shenzhen where
the change in $\mathbf{H}$ is larger than the change in $\mathbf{x}$.

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

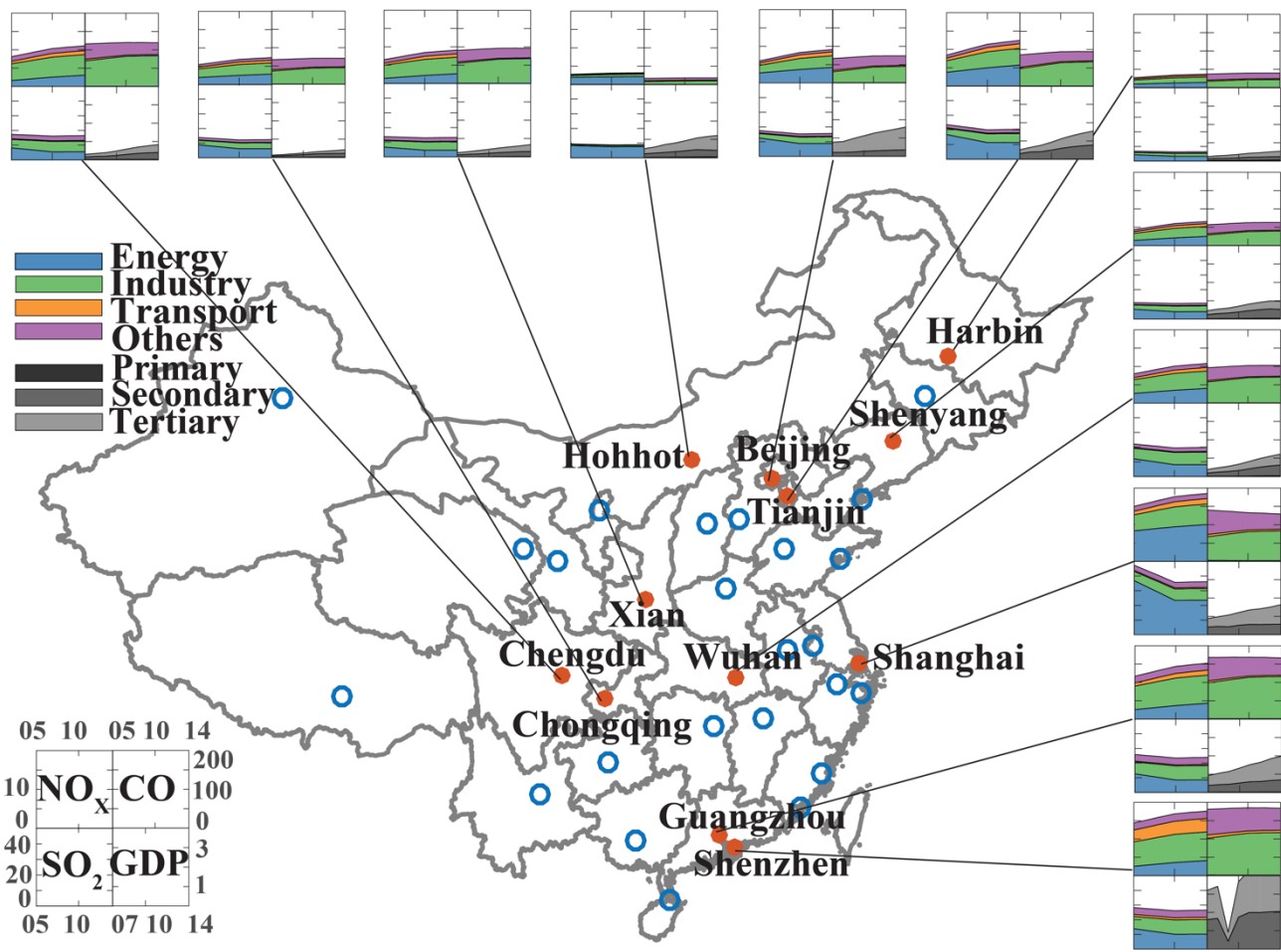

**Figure 1:** Time series (2005-2014) of RCP8.5 combustion-related emissions of $NO_X$ (1st quad), CO (2nd quad) and $SO_2$ (3rd quad) all in units of g/year/m$^2$ and GDP per capita (4th quad) in units of $10^5$RMB/capita/year for each of the 12 select major cities (red dots) in mainland China. The scales of each quadrant are indicated in the legend (lower-left of the map). The total emissions for each combustion product is broken down into 4 major sectors: energy, industry, land transport, and others which is the sum of agriculture, residential and commercial, and waste treatment and disposal). The GDP per capita is also broken down into primary (direct use of natural resources), secondary (industry and manufacturing), and tertiary (service) sectors. Each blue dot corresponds to one of the 36 designated provincial capital and special cities in mainland China.

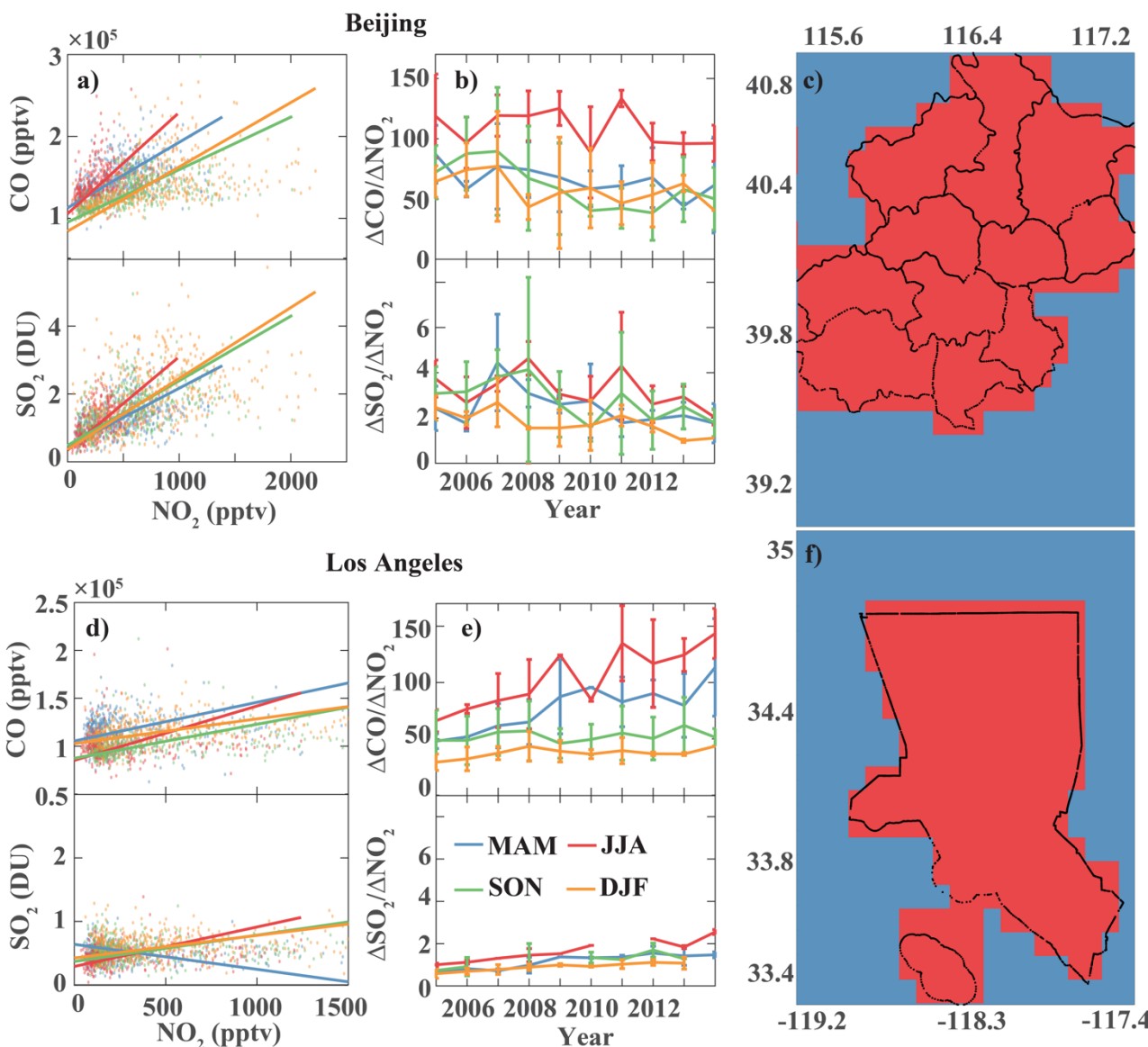

**Figure 2: Spatial regression analysis of satellite retrievals of CO and SO₂ to NO₂ by season (blue: March-May (MAM); red: June-August (JJA); green: September-November (SON); orange: December-February (DJF)). The left column shows an example of scatter plots and linear regression for Beijing (top) and Los Angeles (bottom). The center column corresponds to the changes across 2005 to 2014 on the ratios calculated for a given season. The rightmost column panels show the city domain (2deg x 2deg) with the geopolitical extent of the city of Beijing and Los Angeles.**

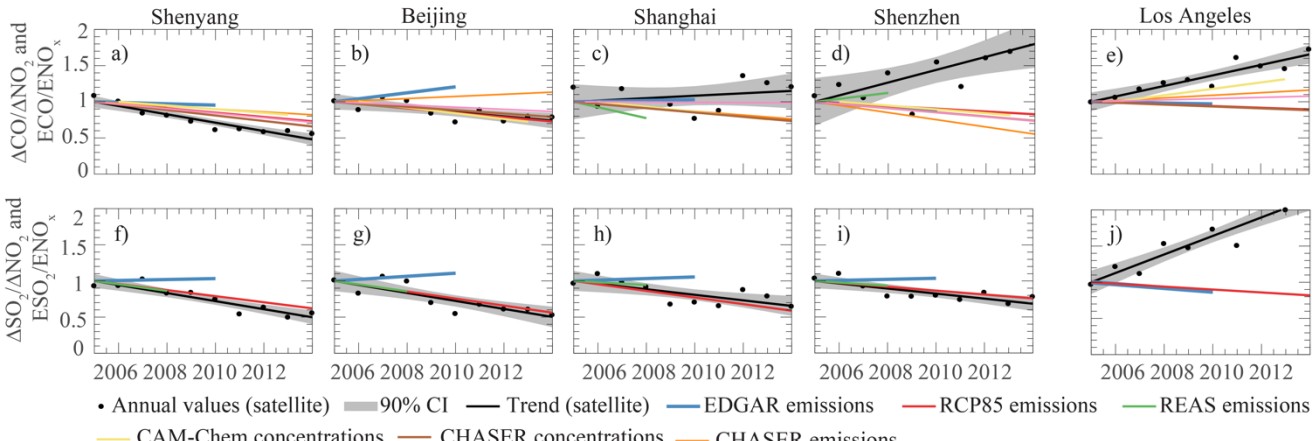

**Figure 3: Changes in annual-mean enhancement ratios (black) from MOPITT and OMI retrievals of CO to $NO_2$ (top) and $SO_2$ to $NO_2$ (bottom) for select cities in China and U.S. relative to year 2005. Its associated emission ratios (($ECO/ENO_X$ and $ESO_2/ENO_X$) from RCP8.5 (red), EDGAR4.2 (blue) and top-down estimate from CHASER (orange) and model-simulated abundance ratios from CHASER (purple) and CAM-Chem (green) chemistry transport models are superimposed. Grey areas are 90% confidence intervals of the linear fit (black lines). The four Chinese cities represent the four classes/levels of urban development across 12 selected cities in China.**

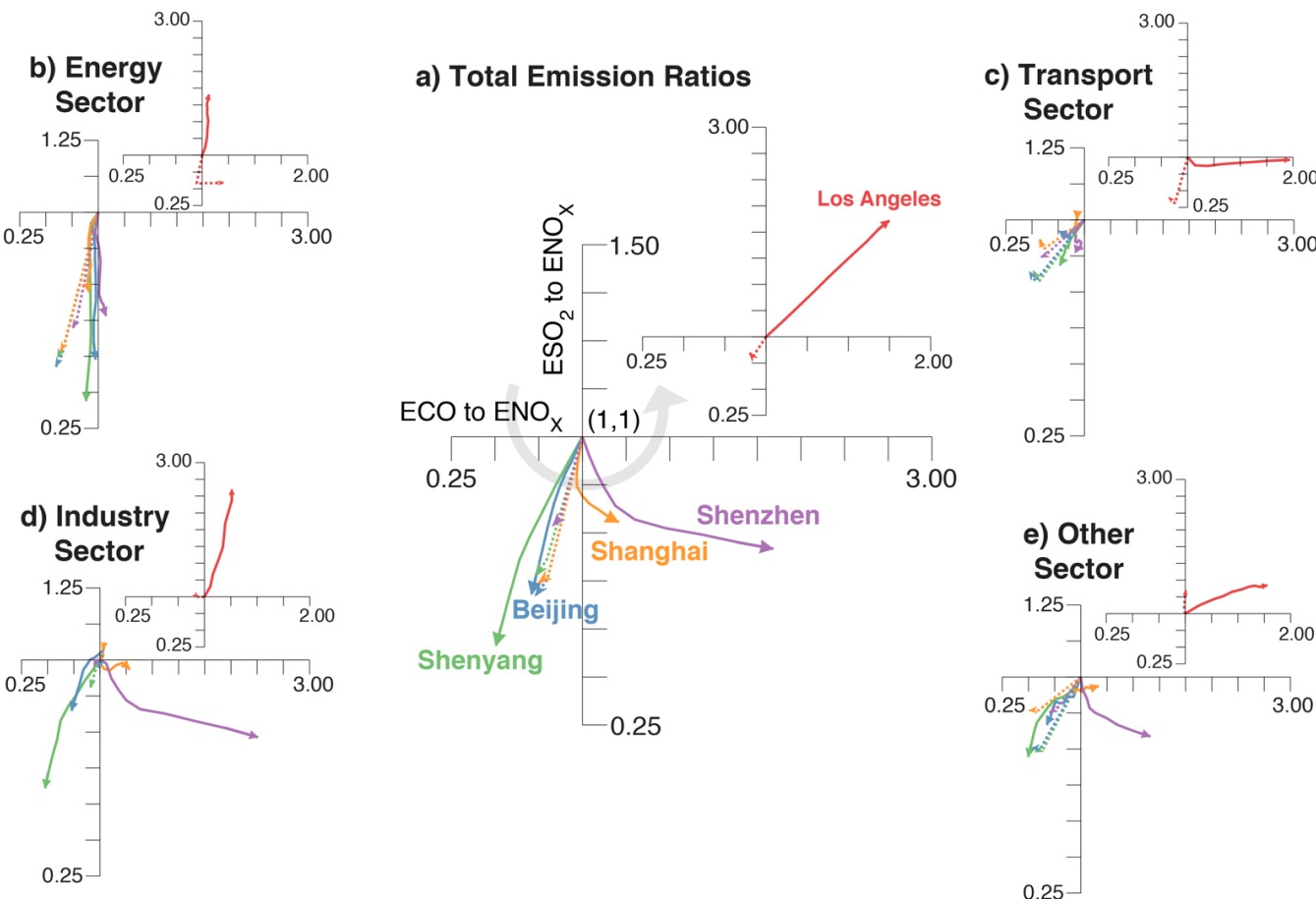

**Figure 4: Joint traces of the annual changes in *a priori* (dotted line) and *a posteriori* (solid line) estimates of E$CO$/E$NO_X$ (x-axis) and E$SO_2$/E$NO_X$ (y-axis) relative to year 2005 for four select Chinese cities (Shenyang: green, Beijing: blue, Shanghai: orange, and Shenzhen: purple) representing four levels of urban development. These traces are presented as line arrows (with origin at x=1, y=1 and endpoint corresponding to year 2005 and 2014, respectively) for total emission ratios (panel a) and four sectoral ratios (panels b to e). Other sector is the sum of mostly residential/commercial along with agriculture, and waste treatment and disposal. The inset for each panel represents the associated traces for Los Angeles, which is added as basis for comparison. The lower-left, lower-right, and upper-right quadrants correspond to decreasing E$CO$/E$NO_X$ and E$SO_2$/E$NO_X$, increasing E$CO$/E$NO_X$ but decreasing E$SO_2$/E$NO_X$, and increasing E$CO$/E$NO_X$ and E$SO_2$/E$NO_X$ relative to year 2005, respectively. The gray semi-circular arrow in panel a) represents our suggested common combustion emission pathway for Chinese cities.**

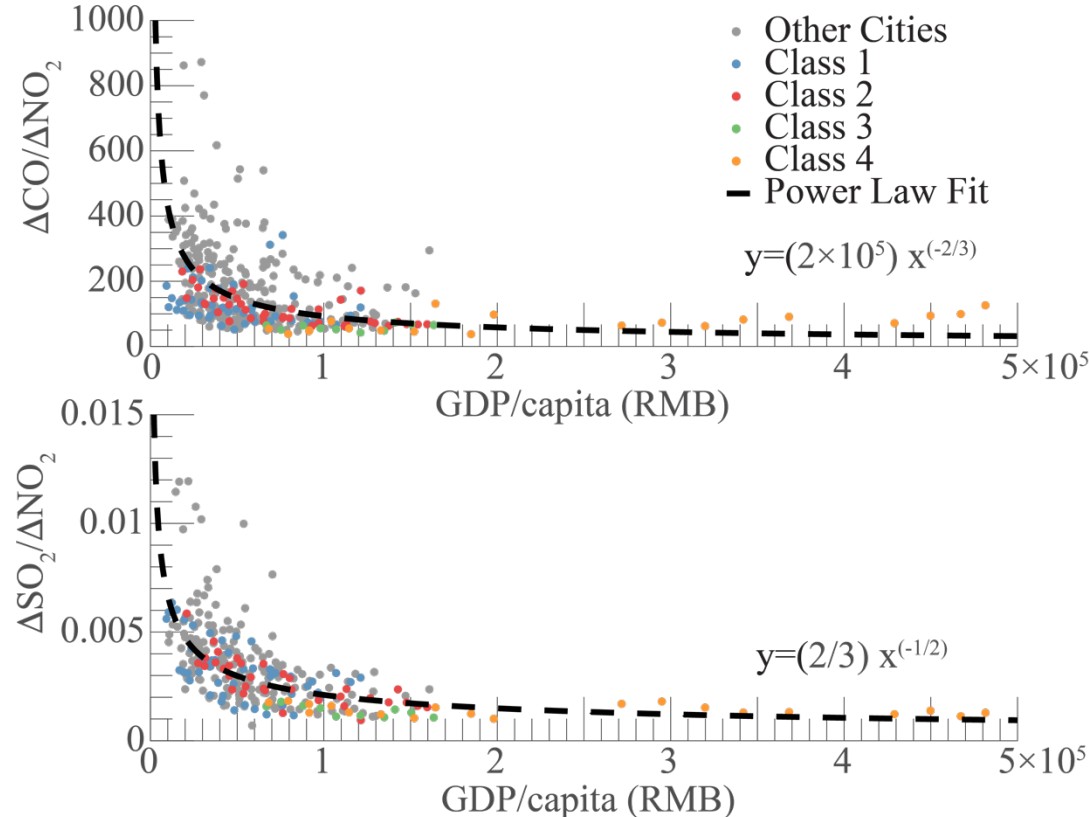

**Figure 5: Annual-mean enhancement ratios (in units of mole/mole) of CO to $NO_2$ (panel a) and $SO_2$ to $NO_2$ (panel b) for all 36 provincial capitals and cities (2005 to 2014) as a function of its corresponding annual GDP/capita (in units of RMB/year/capita). The 12 select cities analyzed in this study are plotted in color, where each color represents four increasing levels or classes of urban development (e.g., Shenyang: Class 1, Beijing: Class 2, Shanghai: Class 3 and Shenzhen: Class 4). The rest of the 36 cities are plotted in gray. Superimposed on panel a) and b) is a fitted curve (black dashed line) based on power-law relationship of the data which is indicated in the plot by its corresponding equation.**

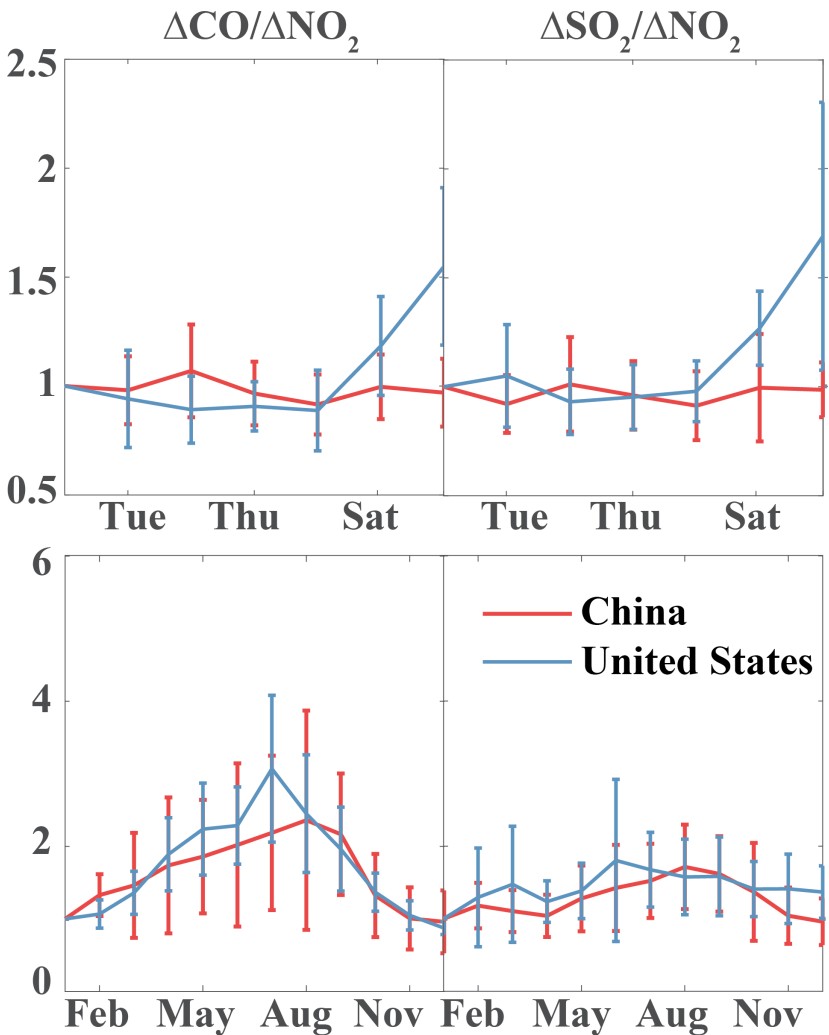

**Figure 6: Weekly (top) and seasonal cycle (bottom) of the satellite-based enhancement ratios averaged for the 12 cities in China (red) and for 8 cities in U.S. (blue). The error bars stand for standard deviation across cities.**

**Table 1. List of satellite products and emission inventories used in this study. All these datasets are re-gridded into 0.1° × 0.1° if the original resolutions are not. This version of CHASER-LETKF does not provide emissions of SO₂.**

| Dataset and Data Availability | Spatial and Temporal Resolution | Relevance to Study & Main Reference |
|---|---|---|
| NASA Terra MOPITT CO version6, L2, TIR/NIR<br>https://www2.acom.ucar.edu/mopitt<br>2000 to present | 22 km × 22 km<br>10:30 AM<br>daily | CO total column<br>(Deeter et al., 2014) |
| Aura/OMI SO2 Total Column 1-orbit L2 v003 NRT<br>https://aura.gsfc.nasa.gov/omi.html<br>2004 - present | 13km × 25 km<br>1:45 PM<br>daily | PBL Column Amount $SO_2$<br>(Krotkov et al., 2006) |
| Dutch OMI $NO_2$ (DOMINO) data product v2.0<br>http://www.temis.nl/airpollution/no2.html<br>2004 to present | 13km × 25 km<br>1:45 PM<br>daily | $NO_2$ trop. column<br>(Boersma et al., 2011) |
| QA4ECV OMI $NO_2$ data product version 1<br>http://temis.nl/qa4ecv/no2col/no2regioomimonth_v2.php<br>2004 to present | 13km × 25 km<br>1:45 PM<br>daily | $NO_2$ trop. column<br>(Boersma et al., 2017) |
| IASI Level 2 FORLI XCO<br>https://navigator.eumetsat.int/product/<br>EO:EUM:DAT:METOP:IASSND02        2007<br>to present | 12km × 12 km<br>9:30 AM<br>daily | CO total column<br>(De Wachter et al., 2012) |
| European Commission EDGAR version 4.3.1<br>http://edgar.jrc.ec.europa.eu/overview.php?v=431<br>1970 to 2010 | 0.1° × 0.1°<br>sectorial<br>annual | CO, $SO_2$, $NO_X$ emissions<br>(Crippa et al., 2016) |
| IIASA  RCPs<br>http://accmip-emis.iek.fz-<br>juelich.de/data/accmip/gridded_netcdf/        1850 to<br>2100 | 0.5° × 0.5°<br>sectorial<br>monthly | CO, $SO_2$, $NO_X$ emissions<br>(Riahi et al., 2011) |
| REAS v2.1                https://www.nies.go.jp/REAS/<br>2000 to 2008 | 0.25° × 0.25°<br>sectorial<br>monthly | CO, $SO_2$, $NO_X$ emissions<br>(Kurokawa et al., 2013) |
| HTAP v2        http://edgar.jrc.ec.europa.eu/htap_v2/<br>2008 and 2010 | 0.1° × 0.1°<br>sectorial<br>monthly | CO, $SO_2$, $NO_X$ emissions<br>(Janssens-Maenhout et al., 2015) |
| CHASER-LETKF<br>https://ebcrpa.jamstec.go.jp/~miyazaki/tcr/<br>2005 to 2014 | 2.8° for longitude and the T42<br>Gaussian grid for latitude<br>daily | CO and $NO_X$ emissions<br>(Miyazaki et al., 2017) |

**Table 2: Summary of Percent Rate of Change for Select Cities in China and United States. Numbers that follow the ± sign are standard errors.**

| | | | | Satellite Observations | | | | |
|---|---|---|---|---|---|---|---|---|
| | | Average GDP (RMB/cap/yr for China and USD/cap/yr for United States) | Annual Rate of Change (RMB/cap/yr for China and USD/cap/yr for United States) | Annual Rate of Change (%/year) | | | | |
| city | class | | | CO | $NO_2$ | $SO_2$ | $\Delta CO/\Delta NO_2$ | $\Delta SO_2/\Delta NO_2$ |
| Shenyang | 1 | 66293 | 8279 | -0.13±0.25 | 5.16±1.40 | -1.92±0.93 | -5.35±0.74 | -6.03±1.02 |
| Xian | 1 | 39594 | 5854 | -0.61±0.22 | 7.45±2.21 | -4.68±1.78 | -4.73±1.44 | -7.55±1.06 |
| Chengdu | 1 | 48722 | 7221 | -1.18±0.52 | 6.93±1.33 | -4.45±1.99 | -4.44±2.25 | -9.58±3.67 |
| Hohhot | 1 | 77744 | 10315 | -0.21±0.23 | 7.49±3.71 | -2.41±1.29 | -3.47±1.78 | -5.68±1.12 |
| Chongqing | 1 | 23706 | 3848 | -0.58±0.41 | 5.65±1.20 | -7.79±2.18 | -3.11±1.49 | -7.67±1.40 |
| Tianjin | 2 | 91503 | 13723 | -0.18±0.28 | 6.09±1.43 | -2.91±1.44 | -3.36±1.61 | -5.46±2.08 |
| Beijing | 2 | 106474 | 11820 | -0.37±0.28 | 3.15±1.70 | -2.04±1.16 | -2.86±1.07 | -5.49±1.42 |
| Harbin | 2 | 35578 | 4079 | 0.07±0.25 | 2.82±1.73 | -0.35±1.13 | -2.69±2.05 | -6.51±1.75 |
| Wuhan | 2 | 67785 | 10940 | -0.70±0.16 | 6.87±1.90 | -4.19±1.53 | -1.83±2.14 | -7.23±1.19 |
| Shanghai | 3 | 115027 | 10809 | -0.34±0.22 | 2.58±1.50 | -4.32±1.23 | 1.40±2.03 | -3.99±1.44 |
| Guangzhou | 4 | 129455 | 14741 | -1.26±0.31 | -3.07±0.76 | -7.00±1.01 | 7.61±6.30 | -4.80±1.24 |
| Shenzhen | 4 | 352018 | 25958 | -1.01±0.20 | -1.77±0.72 | -5.50±1.09 | 8.26±3.08 | -3.40±0.98 |
| Los Angeles | / | 59943 | 215 | -0.47±0.18 | -4.00±0.60 | 0.23±0.29 | 7.34±1.31 | 13.3±1.69 |
| New York | / | 60760 | 516 | -0.44±0.19 | -3.67±0.72 | -1.42±0.54 | 4.98±1.64 | 7.97±1.39 |
| Chicago | / | 57078 | -137 | -0.28±0.18 | -3.30±0.55 | -0.67±0.51 | 7.88±1.84 | 1.48±2.63 |

| | | | | RCP8.5 Emissions | | | | |
|---|---|---|---|---|---|---|---|---|
| | | | | Annual Rate of Change (%/year) | | | | |
| city | class | | | ECO | $ENO_X$ | $ESO_2$ | $ECO/ENO_X$ | $ESO_2/ENO_X$ |
| Shenyang | 1 | | | 1.28±0.17 | 5.85±0.39 | -0.40±0.15 | -2.90±0.24 | -3.94±0.49 |
| Xian | 1 | | | 0.75±0.11 | 4.54±0.31 | -0.47±0.16 | -2.63±0.21 | -3.45±0.43 |
| Chengdu | 1 | | | 0.33±0.07 | 4.10±0.28 | -0.58±0.17 | -2.69±0.22 | -3.32±0.42 |
| Hohhot | 1 | | | 1.14±0.14 | 1.72±0.12 | -0.69±0.14 | -0.50±0.03 | -2.06±0.25 |
| Chongqing | 1 | | | 0.65±0.10 | 3.99±0.27 | -1.21±0.25 | -2.41±0.18 | -3.73±0.49 |
| Tianjin | 2 | | | 1.22±0.17 | 5.38±0.34 | -1.54±0.31 | -2.73±0.20 | -4.49±0.60 |
| Beijing | 2 | | | 1.23±0.18 | 5.83±0.38 | -1.30±0.28 | -2.93±0.22 | -4.50±0.59 |
| Harbin | 2 | | | 0.89±0.11 | 4.07±0.29 | -0.72±0.18 | -2.28±0.18 | -3.41±0.43 |
| Wuhan | 2 | | | 0.74±0.11 | 3.96±0.27 | -1.21±0.25 | -2.33±0.17 | -3.71±0.48 |
| Shanghai | 3 | | | -0.87±0.04 | 2.63±0.19 | -2.73±0.46 | -2.79±0.22 | -4.25±0.60 |
| Guangzhou | 4 | | | -0.06±0.04 | 3.44±0.23 | -0.87±0.20 | -2.63±0.21 | -3.22±0.41 |
| Shenzhen | 4 | | | 0.20±0.06 | 2.54±0.19 | -0.69±0.17 | -1.89±0.14 | -2.58±0.33 |
| Los Angeles | / | | | -5.56±0.30 | -4.91±0.19 | -5.96±0.54 | -1.17±0.10 | -1.95±0.41 |
| New York | / | | | -6.00±0.29 | -5.77±0.25 | -6.60±0.52 | -0.50±0.05 | -1.80±0.33 |
| Chicago | / | | | -5.50±0.32 | -4.99±0.27 | -6.53±0.66 | -0.94±0.05 | -2.89±0.46 |

