# Peer review of "Satellite Data Reveals a Common Combustion Emission Pathway for Major Cities in China"

_Atmospheric Chemistry and Physics, 2018_

## Referee Comment (RC1) · Anonymous Referee #2 · 11 Jan 2019

A very interesting approach is shown to analyse the ratio of CO/NOx and SO2/NOx spatially over megacities and its development over time. The manuscript is basically well-written but it contains some carelessness, which I will mention below.

Page 1, Line 15-19 [Our results ....relative to 2005]: This sentence is very confusing and ambiguously written. A range of ratios is given, but 4 cities are mentioned, it is relative to 2005 and the sentence is ending with an dependent clause. I suggest to split-up this sentence and give a some more explanation.

Page 1, Line 20 [...sectors in Shanghai and Shenzen...]: Only Shanghai and Shenzen are mentioned. What about Shenyang and Beijing?

Page 4, line 21, Page 5, line 14: If you are looking at a 2 x 2 degree area around cities

in China, does this not lead to overlap, for instance, in the case of Guangzhou and Shenzhen.

Page 7, line 18: The results start with Figure 3, while Figure 2 is mentioned later. This is unusual, but moreover I think the storyline of your paper becomes clearer if you start with explaining Figure 2 first.

Page 7, line 24-28: When I compare the numbers of the given ratios with Figure 3, the unit reads %/year instead of %. The rate is in fact an annual rate.

Page 7, line 31 (also on Page 10, line 7): Which four levels of development do you mean ? These development within cities is a very important aspect of the paper, nevertheless the four levels are not discussed nor defined.

Page 8, line 14: Here a reference to Figure 3a is made. However, Figure 3a is not defined while the first subfigure is about Shenyang.

Page 11, line 17: The reference has been forgotten here.

Appendix A, page 15, line 17: "..the fractional contribution of x emission sector f." Change to "..the fractional contribution of emission sector f for species x."

Appendix B: The estimation of H depends on a-prior information because it is an under-determined problem. Can you also give an indication how much information is coming from the measurements and how much of the a-priori ?

Figure 3: - The grey area is very hard to see in this Figure. - It would also be helpful if the underlying data points of the fit are plotted in the Figure as has been done in Figure 2. - The error bars mentioned in the caption are missing.

---

## Referee Comment (RC2) · Anonymous Referee #1 · 24 Jan 2019

This paper reported the decadal changes in the efficiency and cleanness of bulk combustion over large cities in mainland China using satellite observations. The authors have done a lot of works, which are very impressive. It is very interesting to see the temporal variations of SO2/NO2 and CO/NO2. The driving forces of the variations have not been well explained in the text, even though many details are provided. I recommend publishing the paper after reorganizing the parts about driving forces.

General comments:

1. The influence of inter-annual variations due to meteorology. The authors mentioned that analyzing molar ratios rather than absolute molar concentrations contributes to decrease the effects of meteorology. I'm wondering how it works to account for the temporal variation in lifetimes of air pollutants associated with meteorology.

[Figure]

2. The analysis focusing on the differences of SO2/NO2 between the US and China needs substantial improvements. The authors listed the possible reasons for the differences in Page 8. However, no quantitative analysis at urban scale has been performed. For instance, the shares of fuel usage and emissions contributions from different sources for typical cities are expected, which suggest the different emissions characteristic between the US and China. Additionally, the declining SO2/NOx is most likely caused by the de-SO2 procedure in China (Li et al., 2018). The related discussion is missing. The recent reduction in NOx emissions (van der A., et al, 2017; Liu et al., 2016) has not been discussed as well.

3. Too many very lengthy sentences. The authors preferred long sentences through the manuscript. However, those sentences are too long to understand sometimes. I would suggest the authors to go through the text and to simplify some sentences when necessary.

4. section 3.3. This section is trying to explain the driving forces of the trend. It contains many details and the readers may be easily lost. I would recommend the authors to summary the main findings and storyline somewhere in the beginning or at the end of the section, and to reorganize this section based on the summary.

Specific comments:

1. Page 1, line 14, the phrase of "mature satellite instruments sounds not fine. What is the definition of mature? Which instruments are not mature?

2. Page 1, line 31. The English seems to be incorrect.

3. Page 6, line 8. "Here, we treat emissions of these species across the entire extent of the megacity as a point source" As far as my understanding, the authors discarded all the CO and SO2 measurements where there are no NOx measurements. Could you please clarify how do you set up the criteria and why does the criteria make the entire urban areas to a point source?

4. Could you please give the definition of "Combustion Emission Pathway" somewhere?

---

## Author Comment (AC1) · 22 Feb 2019

This paper reported the decadal changes in the efficiency and cleanness of bulk combustion over large cities in mainland China using satellite observations. The authors have done a lot of works, which are very impressive. It is very interesting to see the temporal variations of SO2/NO2 and CO/NO2. The driving forces of the variations have not been well explained in the text, even though many details are provided. I recommend publishing the paper after reorganizing the parts about driving forces.

**Response:** We thank the reviewer for his/her helpful comments on improving the manuscript. We have carefully studied the comments and carried out the revisions accordingly. We believe we have addressed all of them completely. Below is a point-by-point response to the reviewer's comments. We have also provided a copy of the track-change manuscript as well as a clean copy of the revised manuscript.

General comments:
1. The influence of inter-annual variations due to meteorology. The authors mentioned that analyzing molar ratios rather than absolute molar concentrations contributes to decrease the effects of meteorology. I'm wondering how it works to account for the temporal variation in lifetimes of air pollutants associated with meteorology.

**Response:** As the co-emitted species (i.e., CO, $SO_2$, and $NO_2$) are subject to the same meteorological conditions (affecting transport, dilution/mixing, and lifetime), their enhancement ratios are expected to be less sensitive to meteorology compared to the absolute molar concentrations. This is supported by the fact that decadal $\Delta CO/\Delta NO_2$ as well as $\Delta SO_2/\Delta NO_2$ for different seasons have similar trends (Figure 2 of the manuscript). Previous studies have also proven that the ratios compared to the concentrations themselves are relatively immune to changing meteorological conditions, and can provide insights into the magnitude and temporal trends of the emissions (Parrish et al, 2002, 2006, 2009, Silva et al., 2013, Hassler et al. 2016). In addition, they can be directly compared to the corresponding emissions ratios under certain circumstances.

However, we note that even though the ratios derived from satellite observations are relatively less sensitive to meteorology, this methodology cannot eliminate all the impacts from meteorology. The enhancement ratios may be impacted by the meteorological conditions because lifetimes of different air pollutants may respond to meteorological conditions different as the reviewer succinctly pointed out. Nevertheless, we believe such impact should not influence our main conclusions for the following two reasons: (1) Our analysis focuses on decadal trends instead of short-term trends. As shown by previous study, meteorology also plays an important role on relatively short time scales, but meteorology probably plays a lesser role in the longer-term trends (Krotkov et al. 2016); (2) The satellite retrieval samples are taken over the megacities (right above strong emission sources) instead of downwind of the pollution sources, making them more representative of megacity sources.

We thank the reviewer for this helpful comment. We have incorporated the discussion above in the manuscript (Section 3.1, Page 8 Line 27 – Page 9 Line 2 of the track-changed manuscript).

2. The analysis focusing on the differences of SO2/NO2 between the US and China needs substantial improvements. The authors listed the possible reasons for the differences in Page 8. However, no quantitative analysis at urban scale has been performed. For instance, the shares of fuel usage and emissions contributions from different sources for typical cities are expected, which suggest the different emissions characteristic between the US and China. Additionally, the declining SO2/NOx is most likely caused by the de-SO2 procedure in China (Li et al., 2018). The related discussion is missing. The recent reduction in NOx emissions (van der A., et al, 2017; Liu et al., 2016) has not been discussed as well.

**Response:** We have revised and extended the analysis of the differences of $\Delta SO_2/\Delta NO_2$ between the US and China to address the reviewer's concern. We have also included additional references in this part to strengthen the analysis and discussion (Bhattacharya et al., 2015; Hassler et al., 2016; Liu et al., 2016; van der A., et al, 2017; Wu et al., 2017; Li et al., 2018; Sun et al., 2018; Zheng et al., 2018).

The following part:

[revised manuscript text omitted]

3. Too many very lengthy sentences. The authors preferred long sentences through the manuscript. However, those sentences are too long to understand sometimes. I would suggest the authors to go through the text and to simplify some sentences when necessary.

**Response:** We thank the reviewer for point this out. We have gone through the whole manuscript to shorten and/or rephrase the long sentences. Please see the revised manuscript for details.

4. section 3.3. This section is trying to explain the driving forces of the trend. It contains many details and the readers may be easily lost. I would recommend the authors to summary the main findings and storyline somewhere in the beginning or at the end of the section, and to reorganize this section based on the summary.

**Response:** We have divided section 3.3 into three subsections: 3.3.1 Inverse Analysis of the Ratios, 3.3.2 Combustion Emission Pathway, and 3.3.3 Traces in Sectoral Emission Ratios. We have also added the following summary in the beginning of Section 3.3 (Page 11 Line 25 – Line 31 of the track-changed manuscript):

*"We define combustion emission pathway as a trajectory in time of the overall changes in emissions due to combustion with respect to socioeconomic development (e.g., Riahi et al., 2011; Steinberger et al., 2012; Li et al., 2016; Marangoni et al., 2017). In this section, we identify a common combustion emission pathway across these four levels of development and associate them to sectoral changes through inverse analysis. We will briefly describe the inverse analysis of the ratios in section 3.3.1, present our findings on combustion emission pathway in section 3.3.2, and elucidate the driving factors by means of time traces in sectoral emission ratios in section 3.3.3."*

Specific comments:
1. Page 1, line 14, the phrase of "mature satellite instruments sounds not fine. What is the definition of mature? Which instruments are not mature?

**Response:** We have changed the phrase "mature" to "widely used".

**Response:** We have changed the following sentence

"*This is especially problematic since it is in these large cities where anthropogenic activities are most intense, accompanied by immense energy consumption mainly in the form of fossil fuel combustion (Mage et al., 1996; Kennedy et al., 2015)*"

to

"*Anthropogenic activities are most intense in megacities, accompanied by immense energy consumption mainly in the form of fossil fuel combustion (Mage et al., 1996; Kennedy et al., 2015)*".

3. Page 6, line 8. "Here, we treat emissions of these species across the entire extent of the megacity as a point source" As far as my understanding, the authors discarded all the CO and SO2 measurements where there are no NOx measurements. Could you please clarify how do you set up the criteria and why does the criteria make the entire urban areas to a point source?

**Response:** Thank you. In this study, we only use co-located CO and $NO_2$ observations to derive $\Delta CO/\Delta NO_2$, and co-located $SO_2$ and $NO_2$ observations to derive $\Delta SO_2/\Delta NO_2$.

As described in the Section 2.2.1, each city is represented by a $2°\times2°$ area around each city center. And within each city ($2°\times2°$ area), there are 400 of $0.1°\times0.1°$ grids. In another words, during our analyses, a city is represented by 400 grids instead of one single point. And the spatial regression is conducted using 400 grids within each city to obtain enhancement ratios over the city. There is only one enhancement ratio derived from the spatial regression for each city every time. And the enhancement ratios represent bulk characteristics of spatially heterogeneous combustion sources within the megacity. By analyzing the enhancement ratios derived from the spatial regression over the city, we analyze the bulk characteristics of the whole city, which is a complex and mixed signal of all the emission sources and sectors within the city. Therefore, the sentence "*we treat emissions of these species across the entire extent of the megacity as a point source*" only corresponding to the aforementioned methodology of analyzing the bulk characteristics of the whole city using regression ratios. However, we understand the reviewer's concern and realize this sentence could be confusing, so we have deleted it from the manuscript.
In addition, to further understand the bulk characteristics, we do analyze the individual emission sectors contributing to and driving factors of the enhancement ratios that represent the bulk characteristics of the whole city in Section 3.

4. Could you please give the definition of "Combustion Emission Pathway" somewhere?

**Response:** We have added "*We define combustion emission pathway as a trajectory in time of the changes in emissions from combustion in relation to socioeconomic development (e.g., Riahi et al., 2011; Steinberger et al., 2012; Li et al., 2016; Marangoni et al., 2017).*" at the beginning of Section 3.3 (Page 11 Line 25 – Line 27 of the track-changed manuscript). Please also see our response to General Comment 4.

In our case, it includes decadal trend and change of combustion emissions across the megacities in mainland China. Specifically, we found a robust coherent progression of declining-to-growing $\Delta CO/\Delta NO_2$ (-5.4±0.7% to +8.3±3.1%), and slowly-declining $\Delta SO_2/\Delta NO_2$ (-6.0±1.0% to -3.4±1.0%) from Shenyang, Beijing, Shanghai, to Shenzhen relative to 2005. Such progression is well-correlated with economic development, and traces a common emission pathway that resembles evolution of air pollution in more developed cities (Figure 4).

---

## Author Comment (AC2) · 22 Feb 2019

A very interesting approach is shown to analyse the ratio of CO/NOx and SO2/NOx spatially over megacities and its development over time. The manuscript is basically well-written but it contains some carelessness, which I will mention below.

**Response:** We thank the reviewer for his/her helpful comments on improving the manuscript. We have carefully studied the comments and carried out the revisions accordingly. We believe we have addressed all of them completely. Below is a point-by-point response to the reviewer's comments. We have also provided a copy of the track-change manuscript as well as a clean copy of the revised manuscript.

Page 1, Line 15-19 [Our results ....relative to 2005]: This sentence is very confusing and ambiguously written. A range of ratios is given, but 4 cities are mentioned, it is relative to 2005 and the sentence is ending with an dependent clause. I suggest to split-up this sentence and give a some more explanation.

**Response:** Thank you. We have rephrased the sentence to:
"*We present results for four Chinese cities (Shenyang, Beijing, Shanghai, and Shenzhen) representing four levels of urban development. Our results show a robust coherent progression of declining-to-growing $\Delta CO/\Delta NO_2$ relative to 2005 (-5.4±0.7 to +8.3±3.1%), and slowly-declining $\Delta SO2/\Delta NO2$ (-6.0±1.0% to -3.4±1.0%) across the four cities. The coherent progression we found is not evident in the trends of emission ratios reported in Representative Concentration Pathway (RCP8.5) inventory.*"

Page 1, Line 20 [...sectors in Shanghai and Shenzen...]: Only Shanghai and Shenzhen are mentioned. What about Shenyang and Beijing?

**Response:** We have changed the sentence:
"*This progression is likely due to a shift towards cleaner combustion from industrial and residential sectors in Shanghai and Shenzhen, which is presently obfuscated by China's still relatively higher dependence on coal.*"
To

"*This progression is likely due to a shift towards cleaner combustion from industrial and residential sectors in Shanghai and Shenzhen that is not yet seen in Shenyang and Beijing. This overall trend is presently obfuscated by China's still relatively higher dependence on coal.*

Page 4, line 21, Page 5, line 14: If you are looking at a 2 x 2 degree area around cities in China, does this not lead to overlap, for instance, in the case of Guangzhou and Shenzhen.

**Response:** Using 2°×2° area to represent cities does lead to slight overlap over Guangzhou and Shenzhen, Beijing and Tianjin. This does not affect our analyses of emission inventories because we apply geopolitical maps of city boundaries to calculate emissions for each city. This does have an impact on our analyses of satellite observations because we use all the grids in the 2°×2° area to conduct the spatial regression. However, we do not expect the overlap to significantly change

our results because (1) the overlapped area is relatively small; (2) the overlapped cities are sometimes considered together as a whole region because of their similarities and connections (for example, the Jing-Jin-Ji megalopolis and the Pearl River Delta), and (3) the overlapped cities are in the same classes with similar patterns based on our analyses (i.e., Beijing and Tianjin are both in class 2, while Guangzhou and Shenzhen are both in class 4; Table 2).

We appreciate the reviewer for pointing this issue out and have included a similar statement above in Section 2.2.1 (Page 5 Line 8 – Line 17 of the track-changed manuscript).

Page 7, line 18: The results start with Figure 3, while Figure 2 is mentioned later. This is unusual, but moreover I think the storyline of your paper becomes clearer if you start with explaining Figure 2 first.

**Response:** We have restructured Section 3.1 to start with discussion on Figure 2, followed by discussion and analysis of Figure 3. Please see the revised manuscript for details.

Page 7, line 24-28: When I compare the numbers of the given ratios with Figure 3, the unit reads %/year instead of %. The rate is in fact an annual rate.

**Response:** Thank you for pointing it out. The rate is indeed an annual rate and we have changed "%" to "%/year" in the revised manuscript.

Page 7, line 31 (also on Page 10, line 7): Which four levels of development do you mean? These developments within cities is a very important aspect of the paper, nevertheless the four levels are not discussed nor defined.

**Response:** Thank you. The four levels in this study are defined using broad clustering between the average GDP per capita per year and the rate of change in $\Delta CO/\Delta NO_2$ derived from satellite observations. This is shown in Table 2, where a general rule resulting from this analysis would be a classification mainly based on GDP per capita per year, except Harbin and Wuhan.

We have added this statement to Section 3.1 (Page 8 Line 9 – Line 12 of the track-changed manuscript).

Page 8, line 14: Here a reference to Figure 3a is made. However, Figure 3a is not defined while the first subfigure is about Shenyang.

**Response:** Thank you. We have added names for each subfigure in Figure3, and changed "Figure 3a" to "Figure 3e" to refer Los Angeles in the text.

Page 11, line 17: The reference has been forgotten here.

**Response:** We thank the reviewer for pointing it out. We have added Shindell et al. (2011), Zhang et al. (2012), Kheirbek et al. (2014), Yang et al. (2016), and Paulot et al. (2017) to the sentence as references.

Appendix A, page 15, line 17: "..the fractional contribution of x emission sector f."
Change to "..the fractional contribution of emission sector f for species x."

**Response:** We have changed *"... the fractional contribution of x emission sector f."* to *"... the fractional contribution of emission sector f for species x."*

Appendix B: The estimation of H depends on a-prior information because it is an underdetermined problem. Can you also give an indication how much information is coming from the measurements and how much of the a-priori ?

**Response:** Since $\mathbf{H}$ is drawn based on Monte-Carlo sampling, we do not have a diagnostic for the relative contributions of the prior and the data on $\mathbf{H}$. We chose the mean across 100 $\mathbf{H}$ values resulting to estimates of $\mathbf{H}\hat{\mathbf{x}}$ with the lowest RMSEs relative to the data. The changes in $\hat{\mathbf{H}}$ relative to the $\mathbf{H_a}$ can be explored in the sectoral changes shown in Figure 4. This is especially the case for Shanghai and Shenzhen where the change in $\mathbf{H}$ is larger than the change in $\mathbf{x}$.

We have added the statement above in the revised manuscript (Page 18 Line 16 – Line 20 of the track-changed manuscript).

Figure 3: - The grey area is very hard to see in this Figure. - It would also be helpful if the underlying data points of the fit are plotted in the Figure as has been done in Figure 2. - The error bars mentioned in the caption are missing.

**Response:** Thank you. We have adjusted the color of the grey area in Figure 3 (as well as Figure S1) to make it clearer.

We also added underlying data points of the fit of satellite trend in Figure 3 as well as Figure S1. We have deleted the descriptions of the error bars (already had been deleted from the figure), and added descriptions of the grey areas instead.